# DIFFERENTIALLY PRIVATE OPTIMIZATION FOR SMOOTH NON-CONVEX ERM

## ABSTRACT

We develop simple differentially private optimization algorithms that move along directions of (expected) descent to find an approximate second-order necessary solution for non-convex ERM. We use line search, mini-batching, and a two-phase strategy to improve the speed and practicality of the algorithm. Numerical experiments demonstrate the effectiveness of these approaches.

## 1 INTRODUCTION

Privacy protection has become a central issue in machine learning algorithms, and differential privacy (Dwork & Roth, 2014) is a rigorous and popular framework for quantifying privacy. In our paper, we propose a differentially private optimization algorithm that finds an approximate second-order necessary solution for ERM problems. We proposed several techniques to improve the practical performance of the method, including backtracking line search, mini-batching, and a heuristic to avoid the effects of conservative assumptions made in the analysis.

For given $f : \mathbb{R}^d \to \mathbb{R}$, consider the following minimization problem,

$$\min_{w \in \mathbb{R}^d} f(w).$$

We want to find an approximate second-order necessary solution, defined formally as follows.

**Definition 1** (($\epsilon_g, \epsilon_H$)-2NS)**.** For given positive values of $\epsilon_g$ and $\epsilon_H$, We say that $w$ is an ($\epsilon_g, \epsilon_H$)-approximate second-order necessary solution (abbreviated as ($\epsilon_g, \epsilon_H$)-2NS) if

$$\|\nabla f(w)\| \le \epsilon_g, \quad \lambda_{\min}\left(\nabla^2 f(w)\right) \ge -\epsilon_H. \tag{1}$$

We are mostly interested in the case of $\epsilon_g = \alpha$ and $\epsilon_H = \sqrt{M\alpha}$, that is, we seek an ($\alpha, \sqrt{M\alpha}$)-2NS, where $M$ is the Lipschitz constant for $\nabla^2 f$.

We will focus on the empirical risk minimization (ERM) problem, defined as follows,

**Definition 2** (ERM)**.** Given a dataset $D = \{x_1, \ldots, x_n\}$ and a loss function $l(w, x)$, we seek the parameter $w \in \mathbb{R}^d$ that minimizes the empirical risk

$$f(w) = L(w, D) := \frac{1}{n} \sum_{i=1}^{n} l\left(w, x_i\right).$$

ERM is a classical problem in machine learning that has been studied extensively; see, for example Shalev-Shwartz & Ben-David (2014). In this paper, we describe differentially private (DP) techniques for solving ERM.

Previous research on DP algorithms for ERM and optimization has focused mainly on convex loss functions. Recent research on differentially private algorithms for non-convex ERM (Wang et al., 2018a; Wang & Xu, 2019; Zhang et al., 2017) targets an approximate stationary point, which satisfies only the first condition in (1). Wang & Xu (2021) proposes a trust-region type (DP-TR) algorithm that gives an approximate second-order necessary solution for ERM, satisfying both conditions in (1), for particular choices of $\epsilon_g$ and $\epsilon_H$. This work requires the trust-region subproblem to be solved exactly at each iteration, and fixes the radius of the trust region at a small value, akin to a "short step" in a line-search method. An earlier work (Wang et al., 2019) proposed the DP-GD algorithm, which takes short steps in a noisy gradient direction, then sorts through all the iterates so generated to find one that satisfies second-order

necessary conditions. Our work matches the sample complexity bound in DP-GD, which is $O\left(\frac{d}{\alpha^2\sqrt{\rho}}\right)$ for $\rho$-$z$CDP or $O\left(\frac{d\sqrt{\ln(1/\delta)}}{\alpha^2\varepsilon}\right)$ for $(\varepsilon, \delta)$-DP for finding an $(\alpha, \sqrt{M\alpha})$-2NS, and has an iteration complexity of $O(\alpha^{-2})$. Our contributions can be summarized as follows.

- Our algorithm is elementary and is based on a simple (non-private) line-search algorithm for finding an approximate second-order necessary solution. It evaluates second-order information (a noisy Hessian matrix) only when insufficient progress can be made using first-order (gradient) information alone. By contrast, DP-GD uses the (noisy) Hessian only for checking the second-order approximate condition, while DP-TR requires the noisy Hessian to be calculated at every iteration.

- Our algorithm is practical and fast. DP-TR has a slightly better sample complexity bound than our method, depending on $\alpha^{-7/4}$ rather than $\alpha^{-2}$. However, since our analysis is based on the worst case, we can be more aggressive with step sizes (see below). DP-TR requires solving the trust-region subproblem exactly, which is relatively expensive and unnecessary when the gradient is large enough to take a productive step. Experiments demonstrate that our algorithm requires fewer iterations than DP-TR, does less computation on average at each iteration, and thus runs significantly faster than DP-TR. Moreover, we note that the mini-batch version of DP-TR has a sample complexity $O(\alpha^{-2})$, matching the sample complexity of the mini-batch version of our algorithm.

- We use line search and mini-batching to accelerate the algorithm. Differentially private line search algorithms have been proposed by (Chen & Lee, 2020). We use the same sparse vector technique as used by their work, but provide a tighter analysis of the sensitivity of the query for checking sufficient decrease condition. In addition, we provide a rigorous analysis of the guaranteed function decrease with high probability.

- To complement our worst-case analysis, we propose a heuristic that can obtain much more rapid convergence while retaining the guarantees provided by the analysis.

The remainder of the paper is structured as follows. In Section 2, we review basic definitions and properties from differential privacy, and make some assumptions about the function $f$ to be optimized. In Section 3, we describe our algorithm and its analysis. We will discuss the basic short step version of the algorithm in Section 3.1, and an extension to a practical line search method in Section 3.2. A mini-batch adaptation of the algorithm is described in Section 3.3. In Section 4, we present experimental results and demonstrate the effectiveness of our algorithms.

## 2 Preliminaries

We use several variants of DP for the need of our analysis, including $(\varepsilon, \delta)$-DP (Dwork & Roth, 2014), $(\alpha, \epsilon)$-RDP (Mironov, 2017), and $z$CDP (Bun & Steinke, 2016). We review their definitions and properties in Appendix A.

We make the following assumptions about the smoothness of the objective function $f$.

**Assumption 1.** We assume $f$ is lower bounded by $\underline{f}$. Assume further that $f$ is $G$-smooth and has $M$-Lipschitz Hessian, that is, for all $w_1, w_2 \in \text{dom}(f)$,

$$\|\nabla f(w_1) - \nabla f(w_2)\| \le G\|w_1 - w_2\|, \tag{2}$$

$$\|\nabla^2 f(w_1) - \nabla^2 f(w_2)\| \le M\|w_1 - w_2\|, \tag{3}$$

where $\|\cdot\|$ denotes the vector 2-norm and the matrix 2-norm respectively. We use this notation throughout the paper.

For the ERM version of $f$ (see Definition 2), we make additional assumptions.

**Assumption 2.** For the ERM setting, we assume the loss function is $l(w, x)$ is $G$-smooth and has $M$-Lipschitz Hessian with respect to $w$. Thus $L(w, D)$ (the average loss across $n$ samples) is also $G$-smooth and has $M$-Lipschitz Hessian with respect to $w$. In addition, we assume $l(w, x)$ has bounded function values, gradients, and Hessians. That is, there are constants $B$, $B_g$, and $B_H$ such that for any $w, x$ we have,

$$0 \le l(w, x) \le B, \quad \|\nabla_w l(w, x)\| \le B_g, \quad \|\nabla_w^2 l(w, x)\| \le B_H.$$

As a consequence, the $l^2$ sensitivity of $L(w, D)$ and $\nabla L(w, D)$ is bounded by $B/n$ and $2B_g/n$ respectively. We have

$$\|\nabla^2 L(w, D) - \nabla^2 L(w, D')\|_F \le \sqrt{d}\,\|\nabla^2 L(w, D) - \nabla^2 L(w, D')\| \le \frac{2B_H\sqrt{d}}{n}.$$

To simplify notation, we define $g(w) := \nabla f(w)$ and $H(w) := \nabla^2 f(w)$. From the definition (20) of $\ell_2$-sensitivity, we have that the sensitivities of $f$, $g$, and $H$ are

$$\Delta_f = \frac{B}{n}, \quad \Delta_g = 2\frac{B_g}{n}, \quad \Delta_H = \frac{2B_H\sqrt{d}}{n}. \tag{4}$$

## 3 MAIN RESULTS

Our algorithmic starting point is the elementary algorithm described in Wright & Recht (2022, Chapter 3.6) that has convergence guarantees to points that satisfy approximate second-order necessary conditions. For simplicity, we use the following notation to describe and analyze the method:

$$f_k := f(w_k), \quad g_k := g(w_k) = \nabla f(w_k), \quad H_k := H(w_k) = \nabla^2 f(w_k).$$

We employ the Gaussian mechanism to perturb gradients and Hessians, and denote

$$\tilde{g}_k = g_k + \varepsilon_k, \quad \tilde{H}_k = H_k + E_k,$$

where $\varepsilon_k \sim \mathcal{N}(0, \Delta_g^2\sigma_g^2 I_d)$ for some chosen parameter $\sigma_g$ and $E_k$ is a symmetric matrix in which each entry on and above its diagonal is i.i.d. as $\mathcal{N}\left(0, \Delta_H^2\sigma_H^2\right)$, for some chosen value of $\sigma_H$. Let $\tilde{\lambda}_k$ denote the minimum eigenvalue of $\tilde{H}_k$ with $\tilde{p}_k$ the corresponding eigenvector, with sign and norm chosen to satisfy

$$\|\tilde{p}_k\| = 1 \quad \text{and} \quad (\tilde{p}_k)^T\tilde{g}_k \le 0. \tag{5}$$

Algorithm 1 specifies the general form of our optimization algorithm. We will discuss two strategies — a "short step" strategy and one based on backtracking line search — to choose the step sizes $\gamma_{k,g}$ and $\gamma_{k,H}$ to be taken along the directions $\tilde{g}_k$ and $\tilde{p}_k$, respectively. For each variant, we define a quantity MIN_DEC to be the minimum decrease, and use it together with a specific lower bound on $f$ to define an upper bound $T$ of the required number of iterations. In each iteration, we take a step in the negative of the perturbed gradient direction $\tilde{g}_k$ if $\|\tilde{g}_k\| > \epsilon_g$. Otherwise, we check the minimum eigenvalue $\tilde{\lambda}_k$ of the perturbed Hessian $\tilde{H}_k$. If $\tilde{\lambda}_k < -\epsilon_H$, we take a step along the direction $\tilde{p}_k$. In the remaining case, we have $\|\tilde{g}_k\| \le \epsilon_g$ and $\tilde{\lambda}_k \ge -\epsilon_H$, so the approximate second-order necessary conditions are satisfied and we output the current iterate $w_k$ as a 2NS solution.

The quantities $\sigma_f$, $\sigma_g$, $\sigma_H$ determine the amount of noise added to function, gradient, and Hessian evaluations, respectively, with the goal of preserving privacy via Gaussian Mechanism. We can target a certain privacy level for the overall algorithm ($\rho$ in $\rho$-zCDP, for example), find an upper bound on the number of iterations required by whatever variant of Algorithm 1 we are using, and then choose $\sigma_f$, $\sigma_g$, and $\sigma_H$ to ensure this level of privacy. Conversely, we can choose positive values for $\sigma_f$, $\sigma_g$, and $\sigma_H$ and then determine what level of privacy can be ensured by this choice. We can keep track of the privacy leakage as the algorithm progresses, leading to the possibility of adaptive schemes for choosing the $\sigma$'s.)

### 3.1 SHORT STEP

In the short step version of the algorithm, we make choices for the step sizes that are independent of $k$:

$$\gamma_{k,g} \equiv \frac{1}{G}, \quad \gamma_{k,H} \equiv \frac{2|\tilde{\lambda}_k|}{M}. \tag{7}$$

The choices of MIN_DEC and the noise parameters $\sigma_f$, $\sigma_g$, and $\sigma_H$ are discussed in the following results.

First, we discuss the privacy guarantee and its relationship to the noise variances and the number of iterations.

**Theorem 1.** *Let the noise variances $\sigma_f$, $\sigma_g$, $\sigma_H$ be given. Suppose a run of Algorithm 1 takes $k_g$ gradient steps and $k_H$ negative curvature steps. Then the run is $\rho$-zCDP where $\rho = \frac{1}{2}\left(\frac{1}{\sigma_f^2} + \frac{k_g+k_H}{\sigma_g^2} + \frac{k_H}{\sigma_H^2}\right)$.*

*Recall that $T$ is the maximum number of iterations defined in (6). Let $\bar{\rho} = \frac{1}{2}\left(\frac{1}{\sigma_f^2} + \frac{T}{\sigma_g^2} + \frac{T}{\sigma_H^2}\right)$. We always have $\bar{\rho} \ge \rho$, so the algorithm is $\bar{\rho}$-zCDP. Conversely, for given $\rho > 0$ and $\rho_f \in (0, \rho)$, we can choose*

$$\sigma_f^2 = \frac{1}{2\rho_f}, \quad \sigma_g^2 = \sigma_H^2 = \frac{T}{\rho - \rho_f}. \tag{8}$$

*to ensure that the algorithm is $\rho$-zCDP.*

**Algorithm 1** DP Optimization with Second-Order Guarantees
---
**Given:** minimum decrease per iteration MIN_DEC, tolerances $\epsilon_g$ and $\epsilon_H$, noise parameters $\sigma_f$, $\sigma_g$ and $\sigma_H$
Initialize $w^0$ and sample $z \sim \mathcal{N}(0, \Delta_f^2 \sigma_f^2)$
Compute an upper bound of the required number of iterations as follows

$$T = \left\lceil \frac{f(w^0) + |z| - \underline{f}}{\text{MIN\_DEC}} \right\rceil \tag{6}$$

Set $\sigma_g$ and $\sigma_H$ using $T$ (See theorems for details)
**for** $k = 1, 2, \ldots, T$ **do**
  Sample $\varepsilon_k \sim \mathcal{N}\left(0, \Delta_g^2 \sigma_g^2 I_d\right)$
  Compute the perturbed gradient $\tilde{g}_k = g_k + \varepsilon_k$
  **if** $\|\tilde{g}_k\| > \epsilon_g$ **then**
    Choose $\gamma_{k,g}$ and set $w_{k+1} \leftarrow w_k - \gamma_{k,g} \tilde{g}_k$           ▷ Gradient step
  **else**
    Sample $E_k$ such that $E_k$ is a $d \times d$ symmetric matrix in which each entry on and above its diagonal is i.i.d. as $\mathcal{N}\left(0, \Delta_H^2 \sigma_H^2\right)$. Compute perturbed Hessian $\tilde{H}_k = H_k + E_k$ and $(\tilde{\lambda}_k, \tilde{p}_k)$
    **if** $\tilde{\lambda}_k < -\epsilon_H$ **then**
      Choose $\gamma_{k,H} > 0$ and set $w_{k+1} \leftarrow w_k + \gamma_{k,H} \tilde{p}_k$           ▷ Negative curvature step
    **else**
      **return** $w_k$
    **end if**
  **end if**
**end for**
---

*Proof.* The proof follows directly from the $z$CDP guarantee for the Gaussian mechanism combined with postprocessing and composition of $z$CDP. $\qquad\square$

**Remark.** In our algorithm, the actual noise is scaled by the corresponding sensitivity $\Delta$ defined in (4). We do the same for later algorithms. In practice, we expect most steps to be gradient steps, so $\bar{\rho}$ is an overestimate of the actual privacy level $\rho$. We can be more aggressive in choosing the noise variances. We discuss a two-phase approach in Section 3.4.

We now discuss guarantees of the output of Algorithm 1. First, we analyze MIN_DEC in each short step.

**Lemma 2.** *With the short step size choices* (7)*, if the noise satisfies the following conditions for some positive constants $c$, $c_1$, and $c_2$ such that $c_1 < \frac{1}{2}$ and $c_2 + c < \frac{1}{3}$,*

$$\|\varepsilon_k\| \leq \min\left(c_1 \epsilon_g, \frac{c_2}{M} \epsilon_H^2\right), \tag{9a}$$

$$\|E_k\| \leq c\, \epsilon_H, \tag{9b}$$

*then the amount of decrease in each step is at least*

$$\text{MIN\_DEC} = \min\left(\frac{1 - 2c_1}{2G} \epsilon_g^2, \; 2\left(\frac{1}{3} - c_2 - c\right) \frac{\epsilon_H^3}{M^2}\right). \tag{10}$$

*The true gradient and true minimum eigenvalue of the Hessian satisfy the following,*

$$\|g_k\| \leq (1 + c_1) \|\tilde{g}_k\|, \quad \lambda_k > -(1 + c)|\tilde{\lambda}_k|. \tag{11}$$

**Remark.** The constants $c$, $c_1$, and $c_2$ in the conditions (9) above control how accurate our noisy gradient and Hessian estimates are. MIN_DEC is smaller when we choose smaller constants. We will also have a tighter solution as demonstrated in the corollary below. However, we need smaller noise to satisfy the conditions (9), which in turn translates to a larger required sample size $n$ for our ERM problem, as we will see in Theorem 4.

**Corollary 3.** *Assuming the noise satisfies* (9) *at each iteration, the short step version (using* (7)*,* (10)*) of the algorithm will output a $((1 + c_1)\epsilon_g, (1 + c)\epsilon_H)$-2NS.*

With the results above, we now analyze the guarantees of the fixed step-size algorithm under the ERM setting.

**Theorem 4** (Sample complexity of the short step algorithm). *Consider the ERM setting. Suppose that the number of samples $n$ satisfies $n \geq n_{\min}$, where*

$$n_{\min} := \max \left( \frac{\sqrt{2d} B_g \sigma_g \log \frac{T}{\zeta}}{\min \left( c_1 \epsilon_g, \frac{c_2}{M} \epsilon_H^2 \right)}, \frac{C \sqrt{d} B_H \sigma_H \log \frac{T}{\zeta}}{c \epsilon_H} \right).$$

*With probability at least $\{(1 - \frac{\zeta}{T})(1 - C \exp(-cCd))\}^T$ where $c$ and $C$ are universal constants in Lemma D.7, the output of the short step version (using (7),(10)) of the algorithm is a $((1 + c_1)\epsilon_g, (1 + c)\epsilon_H)$-2NS.*

*With the choice of $\sigma$'s in (8) using $\rho_f = c_f \rho$ for $c_f \in (0, 1)$, hiding logarithmic terms and constants, the asymptotic dependence of $n_{\min}$ on $(\epsilon_g, \epsilon_H)$, $\rho$ and $d$, is*

$$n_{\min} = \frac{d}{\sqrt{\rho}} \tilde{O} \left( \max \left( \epsilon_g^{-2}, \epsilon_g^{-1} \epsilon_H^{-2}, \epsilon_H^{-7/2} \right) \right). \tag{12}$$

*When $(\epsilon_g, \epsilon_H) = (\alpha, \sqrt{M\alpha})$, the dependence simplifies to $\frac{d}{\sqrt{\rho}} \tilde{O}(\alpha^{-2})$.*

**Remark.** When the conditions (9) do not hold, the algorithm could fail to converge to a $((1 + c_1)\epsilon_g, (1 + c)\epsilon_H)$-2NS. First, the noise in the perturbed gradient and Hessian can be so large that the step is not a descent direction. Second, due to the noise, we may terminate early or fail to terminate timely when checking the approximate second-order conditions. If the noise is not excessive, the solution is still acceptable since the noisy evaluations satisfy the termination conditions.

## 3.2 LINE SEARCH ALGORITHM

Instead of using a conservative fixed step size, we can do a line search using backtracking. The backtracking line search requires an initial value $\gamma_0$, a decrease parameter $\beta \in (0, 1)$ for the step size, and constants $c_g \in (0, 1 - c_1)$, $c_H \in (0, 1 - c - \sqrt{\frac{8}{3} c_2})$ that determine the amount of decrease we need. Each line search tries in succession the values $\gamma_0, \beta\gamma_0, \beta^2\gamma_0, \ldots$, until a value is found that satisfies a sufficient decrease condition. For gradient steps, the condition is

$$f(w - \gamma\tilde{g}) \leq f(w) - c_g \gamma \|\tilde{g}\|^2, \tag{SD1}$$

while for negative curvature steps it is

$$f(w + \gamma\tilde{p}) \leq f(w) - \frac{1}{2} c_H \gamma^2 |\tilde{\lambda}|. \tag{SD2}$$

To make line search differentially private, we use the *sparse vector technique* from (Dwork & Roth, 2014). We define queries according to (SD1) and (SD2):

$$q_g(\gamma, w) = f(w) - f(w - \gamma\tilde{g}) - c_g \gamma \|\tilde{g}\|^2, \tag{13a}$$

$$q_H(\gamma, w) = f(w) - f(w + \gamma\tilde{p}) - \frac{1}{2} c_H \gamma^2 |\tilde{\lambda}|, \tag{13b}$$

whose nonnegativity is equivalent to each of the sufficient decrease conditions.

Algorithm 2 specifies the differentially private line search algorithm using SVT, which is adapted from `AboveThreshold` algorithm (Dwork & Roth, 2014).

By satisfying the sufficient decrease conditions, we try to get a more substantial improvement in the function value than for the short step algorithm. As a fallback strategy, we use step sizes similar to the short step values (differing only by a constant factor) if the line search fails, yielding a similar decrease to the short-step case. We state the complete algorithm enhanced with line search in Algorithm 3. In the algorithm, we compute the fall back step size $\bar{\gamma}$ and use a multiplier $b$ ($b > 1$) of them as the initial step size $b\bar{\gamma}$ for the line search. We compute the query sensitivity $\Delta_q$ accordingly and call the private line search subroutine to find a step size $\gamma$ that satisfies the sufficient decrease conditions.

We have the following privacy guarantees.

**Theorem 5.** *Suppose that $\sigma_f$, $\sigma_g$, $\sigma_H$, and $\lambda$ are given. Suppose an actual run of the line search algorithm takes $k_g$ gradient steps and $k_H$ negative curvature steps. The run is $\rho$-zCDP where $\rho = \frac{1}{2} \left( \frac{1}{\sigma_f^2} + \frac{k_g + k_H}{\sigma_g^2} + \frac{k_H}{\sigma_H^2} + \frac{k_g + k_H}{\lambda^2} \right).$*

---

**Algorithm 2** Private backtracking line search using SVT

---

**Given:** query $q$ and its sensitivity $\Delta_q$, initial step size multiplier $b$, fall back step size $\bar{\gamma}$, decrease parameter $\beta$, privacy parameter $\lambda$

**function** DP-LINESEARCH($q, \Delta_q, \gamma^{\text{init}}, \bar{\gamma}, \beta, \lambda$)

    Initialize $\gamma \leftarrow \gamma^{\text{init}}$.

    Sample $\xi \sim \text{Lap}\,(2\lambda\Delta_q)$

    **for** $i = 1, 2, \ldots, i_{\max} = \lfloor \log_\beta \frac{\bar{\gamma}}{\gamma^{\text{init}}} \rfloor + 1$ **do**

        Sample $\nu_i \sim \text{Lap}\,(4\lambda\Delta_q)$

        Evaluate $q_i = q(\gamma)$ and $\tilde{q}_i = q_i + \nu_i$

        **if** $\tilde{q}_i \geq \xi$ **then**

            HALT and output $\gamma$

        **else**

            Set $\gamma \leftarrow \beta\gamma$

        **end if**

    **end for**

    HALT and output $\bar{\gamma}$.

**end function**

---

**Algorithm 3** DP Optimization algorithm with Second-Order Guarantees and Backtracking Line Search

---

**Given:** noise bound parameters $c_1, c_2, c$, sufficient decrease parameters $c_g, c_H$ and initial step size multipliers $b_g$, $b_H$, line search decreasing parameters $\beta_g, \beta_H$, tolerances $\epsilon_g$ and $\epsilon_H$, noise parameters $\sigma_f, \sigma_g, \sigma_H$ and $\lambda_{SVT}$

Initialize $w^0$, sample $z \sim \mathcal{N}(0, \Delta_f^2 \sigma_f^2)$ and compute MIN_DEC according to (17)

Compute an upper bound of the required number of iterations $T = \left\lceil \frac{f(w_0) + |z| - \underline{f}}{\text{MIN\_DEC}} \right\rceil$

Set $\bar{\gamma}_g \leftarrow 2\left(1 - c_1 - c_g\right)/G$

**for** $k \leftarrow 1, 2, \ldots, T$ **do**

    Sample $\varepsilon_k \sim \mathcal{N}\left(0, \Delta_g^2 \sigma_g^2 I_d\right)$

    Compute the perturbed gradient $\tilde{g}_k = g_k + \varepsilon_k$

    **if** $\|\tilde{g}_k\| > \epsilon_g$ **then**

        Define $q_{k,g}(\gamma) = f(w_k) - f(w_k - \gamma\tilde{g}_k) - c_g\gamma\|\tilde{g}_k\|^2$

        Set $\gamma_{k,g}^{\text{init}} \leftarrow b_g\bar{\gamma}_g$, $\Delta_{q_{k,g}} \leftarrow \frac{2}{n}\gamma_{k,g}^{\text{init}} B_g\|\tilde{g}_k\|$         ▷ Line search query sensitivity

        $\gamma_{k,g} \leftarrow$ DP-LINESEARCH($q_{k,g}, \Delta_{q_{k,g}}, \gamma_{k,g}^{\text{init}}, \bar{\gamma}_g, \beta_g, \lambda_{SVT}$)         ▷ Backtracking line search

        $w_{k+1} \leftarrow w_k - \gamma_{k,g}\tilde{g}_k$         ▷ Gradient step

    **else**

        Sample $E_k$ such that $E_k$ is a $d \times d$ symmetric matrix in which each entry on and above its diagonal is i.i.d. as $\mathcal{N}\left(0, \Delta_H^2 \sigma_H^2\right)$. Compute perturbed Hessian $\tilde{H}_k = H_k + E_k$

        Compute the minimum eigenvalue of $\tilde{H}_k$ and the corresponding eigenvector $(\tilde{\lambda}_k, \tilde{p}_k)$ satisfying (5).

        **if** $\tilde{\lambda}_k < -\epsilon_H$ **then**

            Define $q_{k,H}(\gamma) = f(w_k) - f(w_k + \gamma\tilde{p}_k) - \frac{1}{2}c_H\gamma^2|\tilde{\lambda}_k|$

            Set $\bar{\gamma}_{k,H} \leftarrow t_2|\tilde{\lambda}_k|/M$, $\gamma_{k,H}^{\text{init}} \leftarrow b_H\bar{\gamma}_{k,H}$, $\Delta_{q_{k,H}} \leftarrow \frac{2}{n}\gamma_{k,H}^{\text{init}} B_g$

            $\gamma_{k,H} \leftarrow$ DP-LINESEARCH($q_H, \Delta_{q_{k_H}}, \gamma_{k,H}^{\text{init}}, \bar{\gamma}_{k,H}, \beta_H, \lambda_{SVT}$)         ▷ Backtracking line search

            $w_{k+1} \leftarrow w_k + \gamma_{k,H}\tilde{p}_k$         ▷ Negative curvature step

        **else**

            **return** $w_k$

        **end if**

    **end if**

**end for**

---

Recall that $T$ is the maximum number of iterations defined in (6). Let $\bar{\rho} = \frac{1}{2}\left(\frac{1}{\sigma_f^2} + \frac{T}{\sigma_g^2} + \frac{T}{\sigma_H^2} + \frac{T}{\lambda^2}\right)$. We always have $\bar{\rho} \geq \rho$, so the algorithm is $\bar{\rho}$-zCDP. Conversely, for given $\rho > 0$ and $\rho_f \in (0, \rho)$, we can choose

$$\sigma_f^2 = \frac{1}{2\rho_f}, \quad \sigma_g^2 = \sigma_H^2 = \lambda^2 = \frac{3T}{2(\rho - \rho_f)}, \tag{14}$$

*to ensure that algorithm is $\rho$-zCDP.*

*Proof.* We know that SVT is $(1/\lambda)$-DP. Thus, it satisfies $(1/(2\lambda^2))$-zCDP. The result follows directly from the zCDP guarantee for the Gaussian mechanism combined with postprocessing and composition of zCDP. □

We now discuss the guarantee of the output of the algorithm. We first derive necessary conditions for sufficient decrease.

**Lemma 6.** *Assume the same bounded noise conditions* (9) *as before. With the choice of sufficient decrease coefficients $c_g \in (0, 1 - c_1), c_H \in (0, 1 - c - \sqrt{\frac{8}{3}c_2})$, let $\bar{\gamma}_g = 2(1 - c_1 - c_g)/G$ and $\bar{\gamma}_H = t_2|\tilde{\lambda}|/M$ as defined in Algorithm 3, the sufficient decrease conditions* (SD1) *and* (SD2) *are satisfied when $\gamma \leq \bar{\gamma}_g$ and $\gamma \in [(t_1/t_2)\bar{\gamma}_H, \bar{\gamma}_H]$, respectively, where $0 < t_1 < t_2$ are solutions to the following quadratic equation (given our choice of $c, c_2, c_H$, real solutions exist),*

$$r(t) := -\frac{1}{6}t^2 + \frac{1}{2}(1 - c - c_H)t - c_2 = 0,$$

*Explicitly, we have*

$$t_1, t_2 = \frac{3}{2}(1 - c - c_H) \pm 3\sqrt{\frac{1}{4}(1 - c - c_H)^2 - \frac{2}{3}c_2}. \tag{15}$$

*In particular, we have $q_g(\bar{\gamma}_g) \geq 0$ and $q_H(\bar{\gamma}_H) \geq 0$.*

We now derive the minimum amount of decrease for each iteration.

**Lemma 7.** *Using DP line search Algorithm 3, assume the same bounded noise conditions* (9) *as before. With the choice of sufficient decrease coefficients $c_g \in (0, 1 - c_1), c_H \in (0, 1 - c - \sqrt{\frac{8}{3}c_2})$, define $\bar{\gamma}_g$ and $\bar{\gamma}_H$ as before. Choose initial step size multipliers $b_g, b_H > 1$ and decrease parameters $\beta_g \in (0, 1), \beta_H \in (t_1/t_2, 1)$. Let $i_{\max} = \lfloor \log_\beta \max(b_g, b_H) \rfloor + 1$. If $n$ satisfies the following:*

$$n \geq 16\lambda \left( \log i_{\max} + \log \frac{T}{\xi} \right) \max \left( 2b_g \frac{B_g}{c_g \epsilon_g}, 4b_H \frac{B_g M}{t_2 c_H \epsilon_H^2} \right), \tag{16}$$

*with probability at least $1 - \xi/T$, the amount of decrease in a single step is at least*

$$\text{MIN\_DEC} = \min \left( \frac{1}{G}(1 - c_1 - c_g)c_g \epsilon_g^2, \frac{1}{4}c_H t_2^2 \frac{\epsilon_H^3}{M^2} \right). \tag{17}$$

With the results above, we can now analyze the guarantees of the line search algorithm under ERM settings.

**Theorem 8** (Sample complexity of the line search algorithm). *Assuming the same conditions as in the previous lemma, with probability at least $\{(1 - \frac{\zeta}{T})(1 - C\exp(-cCd))(1 - \xi/T)\}^T$, suppose the number of samples $n$ satisfies $n \geq n_{\min}$, where*

$$n_{\min} := \max \left( \frac{\sqrt{2d}B_g \sigma_g \log \frac{T}{\zeta}}{\min\left(c_1\epsilon_g, \frac{c_2}{M}\epsilon_H^2\right)}, \frac{C\sqrt{d}B_H \sigma_H \log \frac{T}{\zeta}}{c\,\epsilon_H}, 16\lambda \left( \log i_{\max} + \log(\frac{T}{\xi}) \right) \max \left( 2b_g \frac{B_g}{c_g \epsilon_g}, 4b_H \frac{B_g M}{t_2 c_H \epsilon_H^2} \right) \right). \tag{18}$$

*The output of the algorithm is a $((1 + c_1)\epsilon_g, (1 + c)\epsilon_H)$-2NS. With the choice of $\sigma$'s and $\lambda$ in* (14)*, hiding logarithmic terms and constants, the asymptotic dependence of $n_{\min}$ on $(\epsilon_g, \epsilon_H)$ and $\rho$, is*

$$n_{\min} = \frac{d}{\sqrt{\rho}}\tilde{O}\left( \max\left( \epsilon_g^{-2}, \epsilon_g^{-1}\epsilon_H^{-2}, \epsilon_H^{-7/2} \right) \right). \tag{19}$$

When $(\epsilon_g, \epsilon_H) = (\alpha, \sqrt{M}\alpha)$, the dependence simplifies to $\frac{d}{\sqrt{\rho}}\tilde{O}(\alpha^{-2})$.

*Proof.* The proof is similar to Theorem 4 (see Appendix D.4) using Lemma 17. We have an additional term in our success probability due to the SVT line search step. For the asymptotic bound of $n_{\min}$, we note that MIN\_DEC (17), $T$, $\sigma_g$ and $\sigma_H$ are the same as those of the short step algorithm, up to a constant. The additional requirement (16) for $n$ is $O\left( \frac{\lambda \log T}{\max(\epsilon_g^{-1}, \epsilon_H^{-2})} \right)$. Since we choose $\lambda = \sigma_g$, it is in the same order as the first term inside the max expression of $n_{\min}$ in (18). Thus, the asymptotic bound of $n_{\min}$ is the same as that of the short step algorithm. □

### 3.3 MINI-BATCHING

We can use mini-batching to speed up the algorithm. Specifically, in each iteration $k$, we sample $m$ data points from $D$ without replacement, forming the mini-batch $S_k$. The objective now only computes the average risk over set $S_k$, that is,

$$f_{k,S_k} := \frac{1}{m} \sum_{i \in S_k} \ell(w, x_i).$$

Likewise, we modify all the prior algorithms by evaluating the gradients and Hessians over the mini-batch $S_k$. We show that the sample complexity of the mini-batch version of the algorithm remains $\tilde{O}(\frac{d\sqrt{\ln(1/\delta)}}{\varepsilon\alpha^2})$ when $(\epsilon_g, \epsilon_H) = (\alpha, \sqrt{M\alpha})$ for $(\varepsilon, \delta)$-DP. The details are in the Appendix B.

### 3.4 DISCUSSION: PRACTICAL IMPROVEMENT AND EIGENVALUE COMPUTATION

The estimate of $T$ in (6) is pessimistic, based on MIN_DEC obtained from our worst-case analysis. we now propose a two-phase strategy to speed up the algorithm. In the first phase, we use a fraction (say $3/4$) of the privacy budget to try out a smaller $T$, which corresponds to less noise and quicker convergence. If we are unable to find a desired solution, we fall back to the original algorithm with the remaining privacy budget and use the last iterate as a warm start.

In the actual implementation, we can use Lanczos method to compute the minimum eigenvalue and eigenvector. This will slightly change our analysis. See Appendix C for a discussion. We comment that Lanczos method only requires Hessian-vector product and thus the full Hessian is not required. We can use automatic differentiation or finite differencing to obtain Hessian-vector product efficiently, reducing the time complexity of this step to $O(d)$ from $O(d^2)$.[1]

## 4 EXPERIMENTS

We carry out numerical experiments to demonstrate the performance of our DP optimization algorithms, following similar experimental protocols to (Wang & Xu, 2021). We use Covertype dataset and perform necessary data pre-processing. Details of the dataset and additional experiments can be found in the Appendix E.

Let $x_i$ be the feature vector and $y_i \in \{-1, +1\}$ be the binary label. We investigate the non-convex ERM loss[2]:

$$\min_{w \in \mathbb{R}^p} \frac{1}{n} \sum_{i=1}^{n} \log\left(1 + \exp\left(-y_i \langle x_i, w \rangle\right)\right) + r(w),$$

where $r(w) = \sum_{i=1}^{p} \frac{\lambda w_i^2}{1 + w_i^2}$ is the non-convex regularizer. In our experiments, we choose $\lambda = 10^{-3}$.

We implement our algorithms and DP-TR in Python using PyTorch. [3] To make the results comparable, we modify DP-TR so that it explicitly checks approximate second-order necessary conditions and stop if these conditions are satisfied, in the same way as in our algorithms. We run the experiment under two settings:

1. Finding a loose solution: $\epsilon_g = 0.060$ and $\epsilon_H \approx 0.245$. In this setting, our requirement for the 2NS is loose. This translates to a large sample size $n$ compared to the required sample complexity.
2. Finding a tight solution: $\epsilon_g = 0.030$ and $\epsilon_H \approx 0.173$. In this setting, our requirement for the 2NS is tight. We have a small sample size $n$ compared to the required sample complexity.

For each setting, we pick different levels of privacy budget $\varepsilon$ and run each configuration with five different random seeds. We convert differential privacy schemes to $(\varepsilon, \delta)$-DP when necessary for the comparison. We present the aggregated results in the tables below. In each entry, we report mean $\pm$ standard deviation of the values across five runs. If any of the five runs failed to find a solution, or it found a solution but failed to terminate due to the noise, we report the runtime with $\times$. Here, the runtime is expressed in a unit determined by the Python function `time.perf_counter()`.

In the table, we use acronyms for methods: TR for DP-TR, OPT for our proposed algorithms and 2OPT for their two-phase variants, OPT-LS for our proposed algorithms with line search, and the ones with "-B" use mini-batching.

---

[1] We still evaluate noisy Hessians in our implementation, since there is no optimized support of this in PyTorch.

[2] Upon checking, the loss has Lipschitz gradients and Hessians as long as the feature vector $x_i$'s are bounded.

[3] We implement DP-GD but cannot produce practical results using the algorithmic parameters described in the DP-GD paper.

Table 1: Covertype: finding a loose solution: $(\epsilon_g, \epsilon_H) = (0.060, 0.245)$

| method | $\varepsilon = 0.2$ | | $\varepsilon = 0.6$ | | $\varepsilon = 1.0$ | |
|---|---|---|---|---|---|---|
| | final loss | runtime | loss | runtime | loss | runtime |
| TR | $0.729 \pm 0.028$ | $10.1 \pm 9.9$ | $0.729 \pm 0.026$ | $8.3 \pm 8.6$ | $0.729 \pm 0.026$ | $9.5 \pm 9.1$ |
| TR-B | $0.729 \pm 0.029$ | $2.2 \pm 2.0$ | $0.728 \pm 0.027$ | $2.2 \pm 2.4$ | $0.729 \pm 0.028$ | $2.5 \pm 2.4$ |
| OPT | $0.581 \pm 0.057$ | $\times$ | $0.712 \pm 0.018$ | $0.6 \pm 0.2$ | $0.712 \pm 0.017$ | $0.5 \pm 0.2$ |
| OPT-B | $0.712 \pm 0.018$ | $3.1 \pm 2.9$ | $0.712 \pm 0.018$ | $3.2 \pm 3.0$ | $0.712 \pm 0.018$ | $2.9 \pm 2.9$ |
| OPT-LS | $0.577 \pm 0.032$ | $\times$ | $0.687 \pm 0.028$ | $\mathbf{0.4 \pm 0.1}$ | $0.699 \pm 0.018$ | $\mathbf{0.4 \pm 0.1}$ |
| 2OPT | $0.626 \pm 0.078$ | $\times$ | $0.712 \pm 0.017$ | $0.6 \pm 0.2$ | $0.712 \pm 0.018$ | $0.6 \pm 0.2$ |
| 2OPT-B | $0.712 \pm 0.018$ | $1.4 \pm 0.3$ | $0.712 \pm 0.018$ | $1.4 \pm 0.4$ | $0.712 \pm 0.018$ | $2.0 \pm 1.7$ |
| 2OPT-LS | $0.699 \pm 0.018$ | $\mathbf{0.5 \pm 0.2}$ | $0.699 \pm 0.018$ | $0.5 \pm 0.2$ | $0.699 \pm 0.018$ | $0.5 \pm 0.2$ |

Table 2: Covertype: finding a tight solution: $(\epsilon_g, \epsilon_H) = (0.030, 0.173)$

| method | $\varepsilon = 0.2$ | | $\varepsilon = 0.6$ | | $\varepsilon = 1.0$ | |
|---|---|---|---|---|---|---|
| | final loss | runtime | loss | runtime | loss | runtime |
| TR | $0.516 \pm 0.005$ | $\times$ | $0.607 \pm 0.007$ | $99.6 \pm 32.2$ | $0.607 \pm 0.005$ | $90.8 \pm 21.6$ |
| TR-B | $0.517 \pm 0.005$ | $\times$ | $0.603 \pm 0.005$ | $32.6 \pm 7.9$ | $0.607 \pm 0.003$ | $33.4 \pm 14.4$ |
| OPT | $0.506 \pm 0.001$ | $\times$ | $0.535 \pm 0.015$ | $\times$ | $0.592 \pm 0.003$ | $1.8 \pm 0.5$ |
| OPT-B | $0.597 \pm 0.003$ | $\mathbf{1.3 \pm 0.3}$ | $0.597 \pm 0.003$ | $1.3 \pm 0.2$ | $0.597 \pm 0.003$ | $1.4 \pm 0.3$ |
| OPT-LS | $0.525 \pm 0.009$ | $\times$ | $0.527 \pm 0.009$ | $\times$ | $0.549 \pm 0.006$ | $\times$ |
| 2OPT | $0.502 \pm 0.001$ | $\times$ | $0.513 \pm 0.003$ | $\times$ | $0.519 \pm 0.003$ | $\times$ |
| 2OPT-B | $0.597 \pm 0.003$ | $2.1 \pm 0.4$ | $0.597 \pm 0.003$ | $2.3 \pm 0.5$ | $0.597 \pm 0.003$ | $2.3 \pm 0.6$ |
| 2OPT-LS | $0.577 \pm 0.008$ | $2.1 \pm 1.0$ | $0.591 \pm 0.001$ | $\mathbf{0.6 \pm 0.1}$ | $0.591 \pm 0.001$ | $\mathbf{0.8 \pm 0.2}$ |

Experimental results show that for finding a loose solution under high privacy budgets $\varepsilon = 0.6, 1.0$, our short step algorithm OPT outperforms TR, with much less runtime and lower final loss. Under the low privacy budget $\varepsilon = 0.2$, although OPT can fail to terminate with success, we see that the final loss is even lower than TR. The reason is as follows, due to the conservative estimate of the decrease, the per iteration privacy budget is low, so we cannot check 2NS conditions accurately enough due to the noise. In practice, we can stop early and the solution is still acceptable despite the failure of the termination. Heuristics may be employed to spend extra privacy budget to check 2NS conditions.

Line search and mini-batching improve upon the short step algorithm, especially when combined with our two-phase strategy. We remark that similar to OPT, OPT-LS has an even more conservative theoretical minimum decrease. The two-phase strategy, using an aggressive estimate of the decrease, complements line search. We observe that 2OPT-LS performs consistently well across all privacy budget levels and under two settings. Finally, we remark that the number of Hessian evaluations is minimal. See Appendix E.3 for details.

## 5 CONCLUSION

We develop simple differentially private optimization algorithms based on an elementary algorithm for finding an approximate second-order optimal point of a smooth nonconvex function. The proposed algorithms take noisy gradient steps or negative curvature steps based on a noisy Hessian on nonconvex ERM problems. To obtain a method that is more practical than conservative short-step methods, we employ line searches, mini-batching, and a two-phase strategy. We track privacy leakage using $z$CDP (RDP for mini-batching). Our work matches the sample complexity of DP-GD, but with a much simplified analysis. Although DP-TR has a better sample complexity, its mini-batched version has the same complexity as ours. Our algorithms have a significant advantage over DP-TR in terms of runtime. 2OPT-LS, which combines the line search and the two-phase strategy, consistently outperform DP-TR in numerical experiments.

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

# A A QUICK REVIEW OF DIFFERENTIAL PRIVACY

**Definition A.1** (($\varepsilon, \delta$)-DP (Dwork & Roth, 2014)). A randomized algorithm $\mathcal{A}$ is ($\varepsilon, \delta$)-DP if for all neighboring datasets $D, D'$ and for all events $S$ in the output space of $\mathcal{A}$, the following holds:

$$\Pr\left(\mathcal{A}(D) \in S\right) \leq e^{\varepsilon} \Pr\left(\mathcal{A}(D') \in S\right) + \delta.$$

When $\delta = 0$, we say $\mathcal{A}$ is $\varepsilon$-DP. In DP-ERM, we say $D'$ is a neighboring dataset of $D$ if they differ on just one data point, that is, by changing one data point $x_k$ in $D$ to $x'_k$, we obtain dataset $D'$.

Rényi-DP (RDP) was introduced by Mironov as a relaxation of the original DP.

**Definition A.2** (Rényi divergence). For two probability distributions $P$ and $Q$ defined over $\mathcal{R}$, the Rényi divergence of order $\alpha > 1$ is

$$D_\alpha(P\|Q) = \frac{1}{\alpha - 1} \log \mathrm{E}_{w \sim Q} \left( \frac{P(w)}{Q(w)} \right)^\alpha.$$

**Definition A.3** (($\alpha, \epsilon$)-RDP (Mironov, 2017)). A randomized algorithm $\mathcal{M} : \mathcal{D} \to \mathcal{R}$ is ($\alpha, \epsilon$)-RDP if for all neighboring dataset pairs $D, D'$, the following holds

$$D_\alpha(\mathcal{M}(D)\|\mathcal{M}(D')) \leq \epsilon.$$

Another notion of differential privacy is Zero-Concentrated Differential Privacy ($z$CDP), which requires a linear bound for the divergence of all orders.

**Definition A.4** ($z$CDP (Bun & Steinke, 2016)). A randomized algorithm $\mathcal{M} : \mathcal{D} \to \mathcal{R}$ satisfies ($\xi, \rho$)-$z$CDP if for all neighboring dataset pairs $D, D'$ and all $\alpha \in (1, \infty)$, the following holds:

$$D_\alpha(\mathcal{M}(D)\|\mathcal{M}(D')) \leq \xi + \rho\alpha.$$

Equivalently, a randomized algorithm $\mathcal{M}$ satisfies $\rho$-$z$CDP if for all $\alpha \in (1, \infty)$, $\mathcal{M}$ satisfies ($\alpha, \xi + \rho\alpha$)-RDP. If this definition holds for $\xi = 0$, we use the term $\rho$-$z$CDP instead.

RDP and $z$CDP have some properties in common (Mironov, 2017; Bun & Steinke, 2016).

**Proposition A.1** (Composition of RDP). Suppose that $\mathcal{M}_1 : \mathcal{D} \to \mathcal{R}_1$ is ($\alpha, \epsilon_1$)-RDP and $M_2 : \mathcal{R}_1 \times \mathcal{D} \to \mathcal{R}_2$ is ($\alpha, \epsilon_2$)-RDP. Then the mechanism defined as $(X, Y)$, where $X \sim \mathcal{M}_1(D)$ and $Y \sim \mathcal{M}_2(X, D)$ is ($\alpha, \epsilon_1 + \epsilon_2$)-RDP.

**Proposition A.2** (Composition of $z$CDP). Suppose that $\mathcal{M}_1 : \mathcal{D} \to \mathcal{R}_1$ is $\rho_1$-$z$CDP and $M_2 : \mathcal{R}_1 \times \mathcal{D} \to \mathcal{R}_2$ is $\rho_2$-$z$CDP. Then the mechanism defined as $(X, Y)$, where $X \sim \mathcal{M}_1(D)$ and $Y \sim \mathcal{M}_2(X, D)$ is ($\rho_1 + \rho_2$)-$z$CDP.

**Proposition A.3** (Preservation under Postprocessing). Consider the mappings $\mathcal{M} : \mathcal{D} \to \mathcal{R}$ and $g : \mathcal{R} \to \mathcal{R}'$. We observe that $D_\alpha(P\|Q) \geq D_\alpha(g(P)\|g(Q))$ by the analog of the data processing inequality. This shows that if $\mathcal{M}(\cdot)$ is $(\alpha, \epsilon) - \mathrm{RDP}$, so is $g(\mathcal{M}(\cdot))$. Similarly, if $\mathcal{M}(\cdot)$ is ($\rho$)-$z$CDP, so is $g(\mathcal{M}(\cdot))$.

We can convert easily from one notion of differential privacy to another.

**Proposition A.4** (RDP to ($\varepsilon, \delta$)-DP). If $\mathcal{M}$ is an ($\alpha, \epsilon$)-RDP mechanism, then it is $\left(\epsilon + \frac{\log 1/\delta}{\alpha - 1}, \delta\right)$-DP for any $0 < \delta < 1$.

**Proposition A.5** ($\epsilon$-DP to $z$CDP). If $\mathcal{M}$ is an $\varepsilon$-DP mechanism, then it is also ($\frac{1}{2}\varepsilon^2$)-$z$CDP.

**Proposition A.6** ($z$CDP to ($\varepsilon, \delta$)-DP). Suppose that $\mathcal{M} : \mathcal{D} \to \mathcal{R}$ is ($\xi, \rho$)-$z$CDP. Then $M$ is also ($\varepsilon, \delta$)-DP for all $\delta > 0$ and

$$\varepsilon = \xi + \rho + \sqrt{4\rho \log(1/\delta)}.$$

Thus to achieve a ($\varepsilon, \delta$)-DP guarantee for given $\varepsilon$ and $\delta$, it suffices to satisfy ($\xi, \rho$)-$z$CDP with

$$\rho = (\sqrt{\varepsilon - \xi + \log(1/\delta)} - \sqrt{\log(1/\delta)})^2 \approx \frac{(\varepsilon - \xi)^2}{4 \log(1/\delta)}.$$

A common way to achieve differential privacy is to add Gaussian noise to the output.

**Proposition A.7** (Gaussian Mechanism). Given any function $h : \mathcal{X}^n \to \mathbb{R}^d$, the Gaussian Mechanism is defined as:

$$\mathbf{G}_\sigma h(D) = h(D) + N(0, \Delta_h^2 \sigma^2 I_d),$$

where $\Delta_h$ denotes the $\ell_2$-sensitivity of the function $h$, defined as

$$\Delta_h = \sup_{D \sim D'} \|h(D) - h(D')\|. \tag{20}$$

The Gaussian Mechanism $\mathbf{G}_\sigma h$ satisfies $(\alpha, \alpha/(2\sigma^2))$-RDP for all $\alpha \in [1, \infty)$ and thus also satisfies $(1/2\sigma^2)$-zCDP.

We defer results for $(\varepsilon, \delta)$-DP to Appendix B.2.

## B  ANALYSIS OF THE MINI-BATCH ALGORITHMS

For the line search version of the algorithm, we need additional assumptions if we want to check the sufficient decrease conditions using the mini-batch loss. For simplicity, we only consider the short version of the algorithm in this section.

### B.1  RDP ANALYSIS

(In this section, we use $s = m/n$ to denote the sample fraction.) We evaluate the gradient and the Hessian similarly on the mini-batch $S_k$, which we will write as $g_{k,S_k}$, $H_{k,S_k}$ and let $\tilde{g}_{k,S_k}$, $\tilde{H}_{k,S_k}$ be their perturbed versions respectively. The other parts of the algorithm remain unchanged. The sensitivity $\Delta_f, \Delta_g, \Delta_H$ as stated in (4) will be scaled accordingly by replacing $n$ in their denominator by the mini-batch size $|m$.

Let $g_k$ and $H_k$ be the gradient and the Hessian evaluated on the full dataset $D$. We can decompose the deviation of their noisy approximation as follows,

$$\|\tilde{g}_{k,S_k} - g_k\| \leq \|\tilde{g}_{k,S_k} - g_{k,S_k}\| + \|g_{k,S_k} - g_k\|, \tag{21a}$$

$$\left\|\tilde{H}_{k,S_k} - H_k\right\| \leq \left\|\tilde{H}_{k,S_k} - H_{k,S_k}\right\| + \|H_{k,S_k} - H_k\|, \tag{21b}$$

where the first term in the bound is due to the added Gaussian noise, and the second term is due to subsampling. We will bound two terms separately with high probability.

We have the following subsampling concentration results from (Kohler & Lucchi, 2017)

**Lemma B.1** (Gradient deviation bound). *We have with probability at least $1 - \eta$ that*

$$\|g_{k,S_k} - g_k\| \leq 4\sqrt{2}B_g \sqrt{\frac{\log(2d/\eta) + 1/4}{|S_k|}}.$$

**Lemma B.2** (Hessian deviation bound). *We have with probability at least $1 - \eta$ that*

$$\|H_{k,S_k} - H_k\| \leq 4B_H \sqrt{\frac{\log(2d/\eta)}{|S_k|}}.$$

For the subsampling error (21), we use Gaussian concentration results described before in 41 to bound the first term, and subsampling results stated above to bound the second term. For iteration $k$, with probability at least $(1 - \frac{\zeta}{T})(1 - C\exp(-cCd))(1 - \eta/T)^2$, we have that

$$\|\tilde{g}_{k,S_k} - g_k\| \leq \sqrt{2d}\eta_g \sigma_g \log \frac{T}{\zeta} + 4\sqrt{2}B_g \sqrt{\frac{\log(2dT/\eta) + 1/4}{|S_k|}},$$

$$\left\|\tilde{H}_{k,S_k} - H_k\right\| \leq C\sqrt{d}\eta_H \sigma_H + 4B_H \sqrt{\frac{\log(2dT/\eta)}{|S_k|}}. \tag{22}$$

It suffices to require that each term in the right-hand sides of the bound above is bounded by $1/2$ of the corresponding term in the right-hand sides of (9), so that we can use a similar analysis.

For deviation due to subsampling we need,

$$4\sqrt{2}B_g\sqrt{\frac{\log(2dT/\eta) + 1/4}{|S_k|}} \leq \frac{1}{2}\min\left(c_1\epsilon_g, \frac{c_2}{M}\epsilon_H^2\right) \tag{23a}$$

$$4B_H\sqrt{\frac{\log(2dT/\eta)}{|S_k|}} \leq \frac{1}{2}c\,\epsilon_H \tag{23b}$$

Rearranging the terms, we have the condition for the size of the mini-batch,

$$|S_k| \geq \max\left(64B_g^2(\log(2dT/\eta) + 1/4)\max\left(c_1^{-2}\epsilon_g^{-2}, \frac{M^2}{c_2^2}\epsilon_H^{-4}\right), 32B_H^2\log(2dT/\eta)c^{-2}\epsilon_H^{-2}\right). \tag{24}$$

The following convergence result is immediate, based on the same analysis as in the full-batch case (cf. Theorem 8).

**Theorem B.3.** *With probability at least $\{(1-\zeta/T)(1-C\exp(-cCd))(1-\eta/T)^2\}^T$, suppose the number of samples $n$ satisfies $n \geq n_{\min}$, where*

$$\begin{aligned}
n_{\min} := \max\Bigg(&\frac{2\sqrt{2d}B_g\sigma_g\log\frac{T}{\zeta}}{\min\left(c_1\epsilon_g, \frac{c_2}{M}\epsilon_H^2\right)}, \frac{2C\sqrt{d}B_H\sigma_H\log\frac{T}{\zeta}}{c\,\epsilon_H}, \\
&s^{-1}64B_g^2(\log(2dT/\eta) + 1/4)\max\left(c_1^{-2}\epsilon_g^{-2}, \frac{M^2}{c_2^2}\epsilon_H^{-4}\right), s^{-1}32B_H^2\log(2dT/\eta)c^{-2}\epsilon_H^{-2}\Bigg).
\end{aligned} \tag{25}$$

*The output of the mini-batch short step algorithm is a $((1+c_1)\epsilon_g, (1+c)\epsilon_H)$-2NS. With the choice of $\sigma$'s in (8), hiding logarithmic terms and constants, the asymptotic dependence of $n_{\min}$ on $(\epsilon_g, \epsilon_H)$ and $\rho$, is*

$$n_{\min} = \frac{d}{\sqrt{\rho}}\tilde{O}\left(\max\left(\epsilon_g^{-2}, \epsilon_g^{-1}\epsilon_H^{-2}, \epsilon_H^{-7/2}\right)\right). \tag{26}$$

We now discuss privacy guarantees. It is impossible to deal with subsampling using $z$-CDP, but under RDP, Wang et al. (2018b) provides a generalized analysis for subsampling.

**Theorem B.4** (RDP for Subsampled Mechanisms). *Given a dataset of $n$ points drawn from a domain $\mathcal{X}$ and a (randomized) mechanism $\mathcal{M}$ that takes an input from $\mathcal{X}^m$ for $m \leq n$, let the randomized algorithm $\mathcal{M} \circ \mathbf{subsample}$ be defined as (1) subsample: subsample without replacement $m$ datapoints of the dataset (with sampling fraction $s = m/n$), and (2) apply $\mathcal{M}$: a randomized algorithm taking the subsampled dataset as the input. For all integers $\alpha \geq 2$, if $\mathcal{M}$ is $(\alpha, \epsilon(\alpha))$-RDP, then this new randomized algorithm $\mathcal{M} \circ \mathbf{subsample}$ obeys $(\alpha, \epsilon'(\alpha))$-RDP where,*

$$\begin{aligned}
\epsilon'(\alpha) \leq \frac{1}{\alpha-1}\log\Bigg(1 + s^2\binom{\alpha}{2}\min\left\{4\left(e^{\epsilon(2)} - 1\right), e^{\epsilon(2)}\min\left\{2, \left(e^{\epsilon(\infty)} - 1\right)^2\right\}\right\} \\
+ \sum_{j=3}^{\alpha}s^j\binom{\alpha}{j}e^{(j-1)\epsilon(j)}\min\left\{2, \left(e^{\epsilon(\infty)} - 1\right)^j\right\}\Bigg). \tag{27}
\end{aligned}$$

For the Gaussian mechanism, we have

$$\epsilon(\alpha) = \frac{\alpha}{2\sigma^2},$$

so $\epsilon_{\mathcal{M}(\infty)} = \infty$ and the bound simplifies to

$$\epsilon'(\alpha) \leq \frac{1}{\alpha-1}\log\left(1 + s^2\binom{\alpha}{2}\min\left\{4\left(e^{1/\sigma^2} - 1\right), 2e^{1/\sigma^2}\right\} + \sum_{j=3}^{\alpha}2s^j\binom{\alpha}{j}e^{(j-1)j/(2\sigma^2)}\right) =: \epsilon'_{\mathcal{N}}(\alpha; \sigma, s),$$

which we denote as $\epsilon'_{\mathcal{N}}(\alpha; \sigma, s)$. When $s$ is small and $\sigma$ is large, we can discard higher-order terms and write the right-hand side as

$$\epsilon'_{\mathcal{N}}(\alpha; \sigma, s) \approx \frac{1}{\alpha-1}\left(s^2\frac{\alpha(\alpha-1)}{2} \cdot 4\frac{1}{\sigma^2}\right) = 2s^2\frac{\alpha^2}{\sigma^2}, \tag{28}$$

where we use the approximation $e^t \approx 1 + t$ for small $t$.

**Theorem B.5.** *Consider the short step version of the algorithm using subsampling. Given the choice of $\sigma_f, \sigma_g, \sigma_H, \lambda$ and sampling fraction $s$. Suppose an actual run of the subsampled algorithm takes $k_g$ gradient steps and $k_H$ negative curvature steps. The run is data-dependent $(\alpha, \epsilon(\alpha))$-RDP where*

$$\epsilon(\alpha) = \frac{\alpha}{2\sigma_f^2} + (k_g + k_H)\epsilon'_{\mathcal{N}}(\alpha; \sigma_g, s) + k_H \epsilon'_{\mathcal{N}}(\alpha; \sigma_H, s).$$

*Let*

$$\bar{\epsilon}(\alpha) = \frac{\alpha}{2\sigma_f^2} + T\epsilon'_{\mathcal{N}}(\alpha; \sigma_g, s) + T\epsilon'_{\mathcal{N}}(\alpha; \sigma_H, s). \tag{29}$$

*We always have $\bar{\epsilon}(\alpha) \geq \epsilon(\alpha)$, so the algorithm is $(\alpha, \bar{\epsilon}(\alpha))$-RDP.*

Given the complexity in the subsampled privacy guarantee, $\epsilon'_{\mathcal{N}}(\alpha; \sigma, s)$, we do not have an explicit formula to set parameters $\sigma_f, \sigma_g, \sigma_H$. However, given $(\varepsilon, \delta)$-DP privacy budget, we can optimize the parameters to meet the privacy guarantee. Recall the conversion from $(\alpha, \epsilon(\alpha))$-RDP to $(\varepsilon_{DP}, \delta_{DP})$-DP, given $\delta$, we solve

$$\varepsilon_{DP}(\epsilon(\cdot)) = \min_{\alpha} \left( \epsilon(\alpha) + \frac{\log 1/\delta_{DP}}{\alpha - 1} \right).$$

So we can optimize the parameters $\sigma_f, \sigma_g, \sigma_H$, such that the following objective is minimized

$$\max\left(\bar{\varepsilon}_{DP} - \varepsilon_{DP}(\bar{\epsilon}(\cdot)), 0\right), \tag{30}$$

where $\bar{\varepsilon}_{DP}$ is the target privacy budget and we replace $\bar{\epsilon}(\cdot)$ with their corresponding versions (29).

## B.2  SAMPLE COMPLEXITY USING $(\varepsilon, \delta)$-DP

Under the $(\varepsilon, \delta)$-DP scheme, subsampling is easier to deal with and we will derive a sample complexity bound. We first introduce some useful results in $(\varepsilon, \delta)$-DP.

**Proposition B.1** (Composition of $(\varepsilon, \delta)$-DP). *Suppose that $\mathcal{M}_1 : \mathcal{D} \to \mathcal{R}_1$ is $(\varepsilon_1, \delta_1)$-DP and $\mathcal{M}_2 : \mathcal{R}_1 \times \mathcal{D} \to \mathcal{R}_2$ is $(\varepsilon_2, \delta_2)$-DP. Then the mechanism defined as $(X, Y)$, where $X \sim \mathcal{M}_1(D)$ and $Y \sim \mathcal{M}_2(X, D)$ is $(\varepsilon_1 + \varepsilon_2, \delta_1 + \delta_2)$-DP.*

**Definition B.1** (Gaussian Mechanism for $(\varepsilon, \delta)$-DP). *Given any function $h : \mathcal{X}^n \to \mathbb{R}^d$, the Gaussian Mechanism is defined as:*

$$\mathbf{G}_\sigma h(D) = h(D) + N(0, \Delta_f^2 \sigma^2 I_d),$$

*where $\Delta_h$ be the $\ell_2$-sensitivity of the function $h$ and $\sigma \geq \frac{\sqrt{2\ln(1.25/\delta)}\Delta_2(f)}{\epsilon}$. Then, the Gaussian Mechanism $\mathbf{G}_\sigma h$ satisfies $(\epsilon, \delta)$-differential privacy.*

**Theorem B.6** (Privacy amplification via subsampling (Balle et al., 2018)). *. Given a dataset of $n$ points drawn from a domain $\mathcal{X}$ and a (randomized) mechanism $\mathcal{M}$ that takes an input from $\mathcal{X}^m$ for $m \leq n$, let the randomized algorithm $\mathcal{M} \circ \mathbf{subsample}$ be defined as: (1) subsample: subsample without replacement $m$ datapoints of the dataset (sampling parameter $s = m/n$), and (2) apply $\mathcal{M}$: a randomized algorithm taking the subsampled dataset as the input. If $\mathcal{M}$ is $(\varepsilon, \delta)$-DP, then $\mathcal{M} \circ \mathbf{subsample}$ is $(\varepsilon', \delta')$-DP, where $\varepsilon' = \log(1 + s(e^\varepsilon - 1) \leq s(e^\varepsilon - 1))$ and $\delta' = s\delta$.*

**Theorem B.7** (Advanced Composition). *For all $\varepsilon_0, \delta_0, \delta_0' \geq 0$, the class of $(\varepsilon_0, \delta_0)$-differentially private mechanisms satisfies $(\varepsilon, k\delta + \delta')$-differential privacy under $k$-fold adaptive composition for:*

$$\varepsilon = \sqrt{2k\ln(1/\delta')}\varepsilon + k\varepsilon(e^\varepsilon - 1).$$

*As a corollary, for $0 < \varepsilon < 1$, it suffices to choose $\epsilon_0 = \frac{\varepsilon}{2\sqrt{2k\log(1/\delta')}}$ to ensure the composition is $(\varepsilon, k\delta + \delta')$-DP. In particular, we can in addition choose $\delta' = \delta/2$ and $\delta_0 = \delta/(2k)$ to satisfy $(\varepsilon, \delta)$-DP.*

We have the following privacy guarantee for the algorithm using sampling without replacement,

**Theorem B.8.** *Consider the short step version of the algorithm using subsampling. Given privacy parameters $\varepsilon, \delta, \varepsilon_f, \delta_f \in (0, 1)$ such that $\varepsilon_f < \varepsilon$ and $\delta_f < \delta$, subsampling parameter $s$. Let $\varepsilon_0 = (\varepsilon - \varepsilon_f)/(8s\sqrt{2T\ln(2/(\delta - \delta_f))})$ and $\delta_0 = (\delta - \delta_f)/(4sT)$, where $T$ is estimated as before in (6). Set $\sigma_f = \frac{\sqrt{2\ln(1.25/\delta_f)}}{\varepsilon_f}$, $\sigma_g = \sigma_H = \frac{\sqrt{2\ln(1.25/\delta_0)}}{\epsilon_0}$. The algorithm is $(\varepsilon, \delta)$-DP.*

*Proof.* By Gaussian mechanism, the step for estimating $T$ is $(\varepsilon_f, \delta_f)$-DP. It suffices to show the remaining steps are $(\varepsilon - \varepsilon_f, \delta - \delta_f)$. Using advanced composition, we only need to show that each iteration is $(4s\varepsilon_0, 2s\delta_0)$-DP.

Consider a single iteration without subsampling. From the usage of Gaussian mechanism and sparse vector technique, we know that computing the perturbed gradient step and the perturbed Hessian step are both $(\varepsilon_0, \delta_0)$-DP, whereas the backtracking line search step is $\varepsilon_0$-DP. By composition, we know that the whole iteration is $(2\varepsilon_0, 2\delta_0)$-DP. Applying the Privacy Amplification Theorem B.6, we know that each iteration using subsampling is $(4s\varepsilon_0, 2s\delta_0)$-DP. $\qquad \square$

Since (as earlier) we expect most steps to be gradient steps, rather than negative curvature steps, we overestimate the privacy leakage.

**Theorem B.9.** *For $c_f \in (0, 1)$, setting $\varepsilon_f = c_f\varepsilon$ and $\delta_f = c_f\delta$, under the choice of parameters $\sigma_g, \sigma_H$ in Theorem B.3, the asymptotic dependence of $n_{\min}$ in Theorem B.3 on $(\epsilon_g, \epsilon_H)$, $(\varepsilon, \delta)$, , is*

$$n_{\min} = \tilde{O}\left(\frac{d\sqrt{\ln(1/\delta)}}{\varepsilon} \max\left(\epsilon_g^{-2}, \epsilon_g^{-1}\epsilon_H^{-2}, \epsilon_H^{-4}\right)\right). \tag{31}$$

*When $(\epsilon_g, \epsilon_H) = (\alpha, \sqrt{M\alpha})$, the dependence simplifies to $\tilde{O}(\frac{d\sqrt{\ln(1/\delta)}}{\varepsilon\alpha^2})$, matching the result in full-batch version of the algorithm by converting $\rho$-zCDP to $(\varepsilon, \delta)$-DP via $\sqrt{\rho} = O(\frac{\varepsilon}{d\sqrt{\ln(1/\delta)}})$ using Proposition A.6.*

*Proof.* As before, we have $\sqrt{T} = O(\max(\epsilon_g, \epsilon_H^{-3/2}))$. After simplification, the order of $\sigma_g$ and $\sigma_H$ is

$$\frac{s}{\varepsilon}\sqrt{T}\sqrt{\ln(2sT/\delta)\ln(1/\delta)} = \tilde{O}\left(\frac{s}{\varepsilon}\max(\epsilon_g, \epsilon_H^{-3/2})\right).$$

The asymptotic dependence of $n_{\min}$ follows by substituting $\sigma_g$ and $\sigma_H$ into (25). $\qquad \square$

## C  COMPUTATION OF THE SMALLEST EIGENVALUE USING LANCZOS METHOD

In our algorithms, we need to compute the smallest eigenvalue of the perturbed Hessian. This can be done effectively using the randomized Lanczos algorithm. We have the following result from (Carmon et al., 2018),

**Lemma C.1.** *Suppose that the Lanczos method is used to estimate the smallest eigenvalue of $H$ starting with a random vector uniformly generated on the unit sphere, where $\|H\| \le M$. For any $\delta \in [0, 1)$, this approach finds the smallest eigenvalue of $H$ to an absolute precision of $\epsilon/2$, together with a corresponding direction $v$, in at most*

$$\min\left\{n, 1 + \left\lceil \frac{1}{2}\ln\left(2.75n/\delta^2\right)\sqrt{\frac{M}{\epsilon}} \right\rceil\right\} \text{ iterations}$$

*with probability at least $1 - \delta$.*

To use Lanczos method in our algorithm, we output an estimate $\tilde{\lambda}$ of $\lambda_{\min}(\tilde{H})$ along with the corresponding eigenvector, provided that $\tilde{\lambda} \le -\epsilon_H/2$. If $\tilde{\lambda} > -\epsilon_H/2$, we declare that $\lambda_{\min}(\tilde{H}) \ge -\epsilon_H$, with an error probability at most $\delta_L$. Our analysis and convergence results still hold with $\epsilon_H/2$ replacing $\epsilon_H$ and adding the success probability of the Lanczos algorithm $1 - \delta_L$ to the product of the success probability in each iteration.

## D  MISSING PROOFS

### D.1  PROOF OF LEMMA 2

**Lemma D.1.** *With the short step size choices (7), if the noise satisfies the following conditions for some positive constants $c$, $c_1$, and $c_2$ such that $c_1 < \frac{1}{2}$ and $c_2 + c < \frac{1}{3}$,*

$$\|\varepsilon_k\| \le \min\left(c_1\epsilon_g, \frac{c_2}{M}\epsilon_H^2\right), \tag{32a}$$

$$\|E_k\| \le c\epsilon_H, \tag{32b}$$

*then the amount of decrease in each step is at least*

$$\text{MIN\_DEC} = \min\left(\frac{1-2c_1}{2G}\epsilon_g^2,\ 2\left(\frac{1}{3}-c_2-c\right)\frac{\epsilon_H^3}{M^2}\right). \tag{33}$$

*The true gradient and true minimum eigenvalue of the Hessian satisfy the following,*

$$\|g_k\| \le (1+c_1)\|\tilde{g}_k\|, \quad \lambda_k > -(1+c)|\tilde{\lambda}_k|. \tag{34}$$

*Proof.* We will use the following two standard bounds, which follow from the smoothness assumptions on $f$:

$$f(w+p) \le f(w) + \nabla f(w)^\top p + \frac{G}{2}\|p\|^2, \tag{35}$$

$$f(w+p) \le f(w) + \nabla f(w)^T p + \frac{1}{2}p^T\nabla^2 f(w)p + \frac{1}{6}M\|p\|^3. \tag{36}$$

For simplicity, we drop the iteration number $k$ in the analysis below.

For gradient steps we have $\|\tilde{g}\| > \epsilon_g$. We write $g = \tilde{g} - \varepsilon$. Using $\|\varepsilon\| \le c_1\epsilon_g < c_1\|\tilde{g}\|$, it follows from (35) that

$$\begin{aligned}
f(w - \gamma_g\tilde{g}) &\le f - \gamma_g(\tilde{g}-\varepsilon)^T\tilde{g} + \frac{G}{2}\gamma_g^2\|\tilde{g}\|^2 \\
&\le f - \frac{1}{G}(\tilde{g}-\varepsilon)^T\tilde{g} + \frac{1}{2G}\|\tilde{g}\|^2 \\
&\le f - \frac{1}{2G}\|\tilde{g}\|^2 + \frac{1}{G}\|\varepsilon\|\|\tilde{g}\| \\
&\le f - \frac{1}{2G}\|\tilde{g}\|^2 + \frac{1}{G}c_1\|\tilde{g}\|^2 \\
&= f - \frac{1}{2G}(1-2c_1)\|\tilde{g}\|^2 \\
&\le f - \frac{1}{2G}(1-2c_1)\varepsilon_g^2,
\end{aligned}$$

while the true gradient satisfies

$$\|g\| \le \|\tilde{g}\| + \|\varepsilon\| \le (1+c_1)\|\tilde{g}\|.$$

When negative curvature steps are taken, we have $\tilde{\lambda} < -\epsilon_H$. By assumption, we have $\|\varepsilon\| \le \frac{c_2}{M}\epsilon_H^2 < \frac{c_2}{M}|\tilde{\lambda}|^2$ and $\|E\| \le c\,\epsilon_H < c|\tilde{\lambda}|$. Recall the definition (5) of $\tilde{p}$ and we write $g = \tilde{g} - \varepsilon$, $\tilde{H} = H - E$. From (36), we have

$$\begin{aligned}
f(w + \gamma_H\tilde{p}) &\le f + \gamma_H g^T\tilde{p} + \frac{1}{2}\gamma_H^2\tilde{p}^T H\tilde{p} + \frac{1}{6}M\gamma_H^3\|\tilde{p}\|^3 \\
&= f + \gamma_H\tilde{g}^T\tilde{p} + \frac{1}{2}\gamma_H^2\tilde{p}^T\tilde{H}\tilde{p} + \frac{1}{6}M\gamma_H^3\|\tilde{p}\|^3 - \gamma_H\varepsilon^T\tilde{p} - \frac{1}{2}\gamma_H^2\tilde{p}^T E\tilde{p} \\
&\le f + \frac{1}{2}\left(\frac{2|\tilde{\lambda}|}{M}\right)^2(-|\tilde{\lambda}|) + \frac{1}{6}M\left(\frac{2|\tilde{\lambda}|}{M}\right)^3 - \frac{2|\tilde{\lambda}|}{M}\varepsilon^T\tilde{p} - \frac{1}{2}\left(\frac{2|\tilde{\lambda}|}{M}\right)^2\tilde{p}^T E\tilde{p} \\
&\le f - \frac{2}{3}\frac{|\tilde{\lambda}|^3}{M^2} + \frac{2|\tilde{\lambda}|}{M}\|\varepsilon\| + \frac{2|\tilde{\lambda}|^2}{M^2}\|E\| \\
&\le f - \left(\frac{2}{3} - 2c_2 - 2c\right)\frac{|\tilde{\lambda}|^3}{M^2} \\
&\le f - \left(\frac{2}{3} - 2c_2 - 2c\right)\frac{\epsilon_H^3}{M^2},
\end{aligned}$$

provided that $c_2 + c < 1/3$. Let $\lambda$ denote the minimum eigenvalue of $H$. It follows from Weyl's Inequality that

$$|\tilde{\lambda} - \lambda| \le \|E\| \le c|\tilde{\lambda}|,$$

and thus,

$$\lambda > \tilde{\lambda} - c|\tilde{\lambda}| \ge -(1+c)|\tilde{\lambda}|.$$

$\square$

**Lemma D.2.** *Assume the same bounded noise conditions (9) as before. With the choice of sufficient decrease coefficients $c_g \in (0, 1 - c_1), c_H \in (0, 1 - c - \sqrt{\frac{8}{3}c_2})$, let $\bar{\gamma}_g = 2(1 - c_1 - c_g)/G$ and $\bar{\gamma}_H = t_2|\tilde{\lambda}|/M$ as defined in Algorithm 3, the sufficient decrease conditions (SD1) and (SD2) are satisfied when $\gamma \le \bar{\gamma}_g$ and $\gamma \in [(t_1/t_2)\bar{\gamma}_H, \bar{\gamma}_H]$, respectively, where $0 < t_1 < t_2$ are solutions to the following quadratic equation (given our choice of $c, c_2, c_H$, real solutions exist),*

$$r(t) := -\frac{1}{6}t^2 + \frac{1}{2}(1 - c - c_H)t - c_2 = 0,$$

*Explicitly, we have*

$$t_1, t_2 = \frac{3}{2}(1 - c - c_H) \pm 3\sqrt{\frac{1}{4}(1 - c - c_H)^2 - \frac{2}{3}c_2}. \tag{37}$$

*In particular, we have $q_g(\bar{\gamma}_g) \ge 0$ and $q_H(\bar{\gamma}_H) \ge 0$.*

*Proof.* The analysis is similar to that of Lemma 2. Again for simplicity we drop iteration indices $k$.

For gradient steps we have $\|\tilde{g}\| > \epsilon_g$. We write $g = \tilde{g} - \varepsilon$. Using $\|\varepsilon\| \le c_1\epsilon_g < c_1\|\tilde{g}\|$, it follows from (35) that

$$f(w - \gamma\tilde{g}) \le f(w) - \gamma(\tilde{g} - \varepsilon)^\top \tilde{g} + \frac{G}{2}\gamma^2\|\tilde{g}\|^2$$

$$\le f(w) - \left(\gamma - \frac{G}{2}\gamma^2\right)\|\tilde{g}\|^2 + \gamma\|\tilde{g}\|\|\varepsilon\|$$

$$\le f(w) - \gamma\left(1 - \frac{G}{2}\gamma - c_1\right)\|\tilde{g}\|^2,$$

It follows by definition of $\bar{\gamma}_g$ that (SD1) holds when $\gamma \le \bar{\gamma}_g$ and

When negative curvature steps are taken, we have $\tilde{\lambda} < -\epsilon_H$. By assumption, we have $\|\varepsilon\| \le \frac{c_2}{M}\epsilon_H^2 < \frac{c_2}{M}|\tilde{\lambda}|^2$ and $\|E\| \le c\,\epsilon_H < c|\tilde{\lambda}|$. Recall the definition (5) of $\tilde{p}$ and we write $g = \tilde{g} - \varepsilon$, $\tilde{H} = H - E$. From (36), we have for $\gamma > 0$ that

$$f(w + \gamma\tilde{p}) \le f(w) + \gamma\tilde{g}^T\tilde{p} + \frac{1}{2}\gamma^2\tilde{p}^T H\tilde{p} + \frac{1}{6}M\gamma^3\|\tilde{p}\|^3 - \gamma\varepsilon^T\tilde{p} - \frac{1}{2}\gamma^2\tilde{p}^T E\tilde{p}$$

$$\le f(w) - \frac{1}{2}\gamma^2|\tilde{\lambda}| + \frac{1}{6}M\gamma^3 + \gamma\|\varepsilon\| + \frac{1}{2}\gamma^2\|E\|$$

$$\le f(w) - \underbrace{\left(\frac{1}{2}\gamma^2(1 - c)|\tilde{\lambda}| - \gamma\frac{c_2}{M}|\tilde{\lambda}|^2 - \frac{1}{6}M\gamma^3\right)}_{g(\gamma)}.$$

By reparameterizing $\gamma = \frac{t|\tilde{\lambda}|}{M}$, we obtain

$$g(\gamma) - \frac{1}{2}c_H\gamma^2|\tilde{\lambda}| = \left(-\frac{1}{6}t^3 + \frac{1}{2}(1 - c - c_H)t^2 - c_2t\right)\frac{|\tilde{\lambda}|^3}{M^2} = t \cdot r(t)\frac{|\tilde{\lambda}|^3}{M^2}.$$

Note that (SD2) holds when $g(\gamma) \ge \frac{1}{2}c_H\gamma^2|\tilde{\lambda}|$. The result follows from the fact that $r(t) \ge 0$ for $t \in [t_1, t_2]$. $\qquad\square$

## D.2 PROOF OF LEMMA 7

**Lemma D.3.** *Using DP line search Algorithm 3, assume the same bounded noise conditions (9) as before. With the choice of sufficient decrease coefficients $c_g \in (0, 1 - c_1), c_H \in (0, 1 - c - \sqrt{\frac{8}{3}c_2})$, define $\bar{\gamma}_g$ and $\bar{\gamma}_H$ as before. Choose initial step size multipliers $b_g, b_H > 1$ and decrease parameters $\beta_g \in (0, 1), \beta_H \in (t_1/t_2, 1)$. Let $i_{\max} = \lfloor \log_\beta \max(b_g, b_H) \rfloor + 1$. If $n$ satisfies the following:*

$$n \ge 16\lambda\left(\log i_{\max} + \log(T/\xi)\right)\max\left(2b_g\frac{B_g}{c_g\epsilon_g}, 4b_H\frac{B_gM}{t_2c_H\epsilon_H^2}\right), \tag{38}$$

*with probability at least $1 - \xi/T$, the amount of decrease in a single step is at least*

$$\text{MIN\_DEC} = \min\left(\frac{1}{G}(1 - c_1 - c_g)c_g\epsilon_g^2, \frac{1}{4}c_H t_2^2 \frac{\epsilon_H^3}{M^2}\right). \tag{39}$$

*Proof.* The analysis is similar to that of Lemma 2. Again for simplicity we drop iteration indices $k$.

For gradient steps we have $\|\tilde{g}\| > \epsilon_g$. We write $g = \tilde{g} - \varepsilon$. Using $\|\varepsilon\| \leq c_1\epsilon_g < c_1\|\tilde{g}\|$, it follows from (35) that

$$f(w - \gamma\tilde{g}) \leq f(w) - \gamma(\tilde{g} - \varepsilon)^\top\tilde{g} + \frac{G}{2}\gamma^2\|\tilde{g}\|^2$$

$$\leq f(w) - \left(\gamma - \frac{G}{2}\gamma^2\right)\|\tilde{g}\|^2 + \gamma\|\tilde{g}\|\|\varepsilon\|$$

$$\leq f(w) - \gamma\left(1 - \frac{G}{2}\gamma - c_1\right)\|\tilde{g}\|^2,$$

It follows by definition of $\bar{\gamma}_g$ that (SD1) holds when $\gamma \leq \bar{\gamma}_g$ and

When negative curvature steps are taken, we have $\tilde{\lambda} < -\epsilon_H$. By assumption, we have $\|\varepsilon\| \leq \frac{c_2}{M}\epsilon_H^2 < \frac{c_2}{M}|\tilde{\lambda}|^2$ and $\|E\| \leq c\,\epsilon_H < c|\tilde{\lambda}|$. Recall the definition (5) of $\tilde{p}$ and we write $g = \tilde{g} - \varepsilon$, $\tilde{H} = H - E$. From (36), we have for $\gamma > 0$ that

$$f(w + \gamma\tilde{p}) \leq f(w) + \gamma\tilde{g}^T\tilde{p} + \frac{1}{2}\gamma^2\tilde{p}^T H\tilde{p} + \frac{1}{6}M\gamma^3\|\tilde{p}\|^3 - \gamma\varepsilon^T\tilde{p} - \frac{1}{2}\gamma^2\tilde{p}^T E\tilde{p}$$

$$\leq f(w) - \frac{1}{2}\gamma^2|\tilde{\lambda}| + \frac{1}{6}M\gamma^3 + \gamma\|\varepsilon\| + \frac{1}{2}\gamma^2\|E\|$$

$$\leq f(w) - \underbrace{\left(\frac{1}{2}\gamma^2(1 - c)|\tilde{\lambda}| - \gamma\frac{c_2}{M}|\tilde{\lambda}|^2 - \frac{1}{6}M\gamma^3\right)}_{g(\gamma)}.$$

By reparameterizing $\gamma = \frac{t|\tilde{\lambda}|}{M}$, we obtain

$$g(\gamma) - \frac{1}{2}c_H\gamma^2|\tilde{\lambda}| = \left(-\frac{1}{6}t^3 + \frac{1}{2}(1 - c - c_H)t^2 - c_2 t\right)\frac{|\tilde{\lambda}|^3}{M^2} = t \cdot r(t)\frac{|\tilde{\lambda}|^3}{M^2}.$$

Note that (SD2) holds when $g(\gamma) \geq \frac{1}{2}c_H\gamma^2|\tilde{\lambda}|$. The result follows from the fact that $r(t) \geq 0$ for $t \in [t_1, t_2]$.

$\square$

### D.3  PROOF OF COROLLARY 3

**Corollary D.4.** *3 Assuming the noise satisfies* (9) *at each iteration, the short step version (using* (7), (10)*) of the algorithm will output a* $((1 + c_1)\epsilon_g, (1 + c)\epsilon_H)$*-2NS.*

*Proof.* From the minimum decrease (10) we just derived, it follows that the algorithm will terminate in $T^*$ iterations, where

$$T^* = \frac{f(w_0) - f^*}{\text{MIN\_DEC}}.$$

Our choice of $T$ in (6) is an upper bound of $T^*$ and thus the algorithm will halt within $T$ iterations. In the iteration $k$ when the algorithm halts, we have $\|\tilde{g}_k\| \leq \epsilon_g$ and $\tilde{\lambda}_k \geq -\epsilon_H$. It follows from (11) that the output is a $((1 + c_1)\epsilon_g, (1 + c)\epsilon_H)$-2NS.

$\square$

### D.4  PROOF OF THEOREM 4

**Theorem D.5** (Sample complexity of the short step algorithm). *Consider the ERM setting. Suppose that the number of samples $n$ satisfies $n \geq n_{\min}$, where*

$$n_{\min} := \max\left(\frac{\sqrt{2d}B_g\sigma_g\log\frac{T}{\zeta}}{\min\left(c_1\epsilon_g, \frac{c_2}{M}\epsilon_H^2\right)}, \frac{C\sqrt{d}B_H\sigma_H\log\frac{T}{\zeta}}{c\,\epsilon_H}\right).$$

With probability at least $\{(1 - \frac{\zeta}{T})(1 - C \exp(-cCd))\}^T$ where $c$ and $C$ are universal constants in Lemma D.7, the output of the short step version (using (7),(10)) of the algorithm is a $((1 + c_1)\epsilon_g, (1 + c)\epsilon_H)$-2NS.

With the choice of $\sigma$'s in (8) using $\rho_f = c_f \rho$ for $c_f \in (0, 1)$, hiding logarithmic terms and constants, the asymptotic dependence of $n_{\min}$ on $(\epsilon_g, \epsilon_H)$, $\rho$ and $d$, is

$$n_{\min} = \frac{d}{\sqrt{\rho}} \tilde{O}\left(\max\left(\epsilon_g^{-2}, \epsilon_g^{-1}\epsilon_H^{-2}, \epsilon_H^{-7/2}\right)\right). \tag{40}$$

When $(\epsilon_g, \epsilon_H) = (\alpha, \sqrt{M\alpha})$, the dependence simplifies to $\frac{d}{\sqrt{\rho}} \tilde{O}(\alpha^{-2})$. Before proving Theorem 4, we introduce two concentration results.

**Lemma D.6** (Gaussian concentration, (Vershynin, 2018)). *For $x \sim \mathcal{N}\left(0, \sigma^2 I_d\right)$, with probability at least $1 - \eta$ for any $1 > \eta > 0$, we have*

$$\|x\| \leq \sqrt{2d}\sigma \log \frac{1}{\eta}.$$

**Lemma D.7** (Upper tail estimate for Wigner ensembles (Tao, 2012, p. 110)). *Let $M = (m_{ij})_{1 \leq i,j \leq d}$ be an $d \times d$ random symmetric matrix. Suppose that the coefficients $m_{ij}$ of $M$ are independent for $j \geq i$, mean zero, and have uniform sub-Gaussian tails. There exist universal constants $C, c > 0$ such that for all $A \geq C$, we have*

$$\mathbf{P}\left(\|M\| > A\sqrt{d}\right) \leq C \exp(-cAd).$$

*Proof.* It follows from concentration results that, in iteration $k$, with probability at least $(1 - \frac{\zeta}{T})(1 - C \exp(-cCd))$, we have

$$\|\varepsilon_k\| \leq \sqrt{2d}\Delta_g \sigma_g \log \frac{T}{\zeta}, \tag{41a}$$

$$\|E_k\| \leq C\sqrt{d}\Delta_H \sigma_H. \tag{41b}$$

We need to find a condition on $n$ that ensures that the right-hand sides are less than the right-hand sides of (9). We substitute for $\Delta_g$ and $\Delta_H$ from (4) and solve for $n_{\min}$ by rearranging the terms. The result then follows from Corollary 3 if the concentration results hold for all iterations.

Now let us calculate the success probability. For each iteration, we have a probability of at least $(1 - \frac{\zeta}{T})(1 - C \exp(-cCd))$ that the concentration results hold (if we do not compute the perturbed Hessian, the probability is higher with at least $1 - \frac{\zeta}{T}$. Using conditional probability, the overall success probability is $\{(1 - \frac{\zeta}{T})(1 - C \exp(-cCd))\}^\tau$ conditioned on the number of iterations $\tau$. Since $\tau \leq T$, the overall success probability is at least $\{(1 - \frac{\zeta}{T})(1 - C \exp(-cCd))\}^T$.

For the second part, recall from (8) that $\sigma_g = \sigma_H = \sqrt{T}/\sqrt{(1 - c_f)\rho}$. With (10) and our choice of $T$ in (6), we have $\sqrt{T} = O(\max(\epsilon_g, \epsilon_H^{-3/2}))$. We obtain the asymptotic bound of $n_{\min}$ by plugging in $\sigma_g$ and $\sigma_H$. $\qquad\square$

# E  EXPERIMENTAL SETTINGS AND ADDITIONAL EXPERIMENTS

We first remark that all our experiments were run on a cluster with a 36-core Intel Xeon Gold 6254 3.1GHz CPU, utilizing 8 CPU cores for each run.

## E.1  DATASETS

The Covertype dataset[4] contains $n = 581012$ data points. Each data point has the form $(x, y)$, where $x$ is a 54-dimensional feature vector (first 10 are dimensions numerical, column $11 - 14$ is the WildernessArea one-hot vector, and last 40 columns are the SoilType one-hot vector), and $y$ being the label, is one of $\{1, 2, \ldots, 7\}$.

---

[4]Data source: UCI Machine Learning Repository https://archive.ics.uci.edu/ml/datasets/covertype

For preprocessing, we normalize the first 10 numerical columns, and keep only those samples for which $y = 1, 2$. The number of samples remaining in this restricted set is $n = 495141$. We recode $y = 2$ to $y = -1$ so that $y \in \{-1, 1\}$.

The IJCNN dataset[5] contatins $n = 4999$ data points.

Each point consists of $(x, y)$, where $x$ is a 22-dimensional feature vector (first 10 are one hot and column $11 - 22$ are numerical, and $y$ being the label is binary. For preprocessing, we normalize the data.

For privacy accounting of RDP, which is used in the mini-batched algorithm, we use the `autodp` package[6].

Below we repeat the same experiment using the IJCNN dataset.

### E.2  IJCNN EXPERIMENT USING LOSS IN SECTION 4

Table 3: IJCNN: finding a loose solution: $(\epsilon_g, \epsilon_H) = (0.040, 0.200)$

| method | $\varepsilon = 0.2$ | | $\varepsilon = 0.6$ | | $\varepsilon = 1.0$ | |
| --- | --- | --- | --- | --- | --- | --- |
| | final loss | runtime | loss | runtime | loss | runtime |
| TR | $0.621 \pm 0.009$ | $\times$ | $0.71 \pm 0.025$ | $1.3 \pm 1.5$ | $0.718 \pm 0.019$ | $0.8 \pm 1.1$ |
| TR-B | $0.622 \pm 0.009$ | $\times$ | $0.718 \pm 0.046$ | $0.4 \pm 0.4$ | $0.72 \pm 0.043$ | $1.3 \pm 1.6$ |
| OPT | $0.603 \pm 0.012$ | $\times$ | $0.644 \pm 0.014$ | $\times$ | $0.702 \pm 0.015$ | $0.1 \pm 0.0$ |
| OPT-B | $0.71 \pm 0.022$ | $2.7 \pm 2.4$ | $0.71 \pm 0.022$ | $2.6 \pm 2.5$ | $0.71 \pm 0.022$ | $2.8 \pm 2.3$ |
| OPT-LS | $0.671 \pm 0.03$ | $\times$ | $0.593 \pm 0.02$ | $\times$ | $0.658 \pm 0.019$ | $\times$ |
| 2OPT | $0.631 \pm 0.052$ | $\times$ | $0.679 \pm 0.048$ | $\times$ | $0.676 \pm 0.056$ | $\times$ |
| 2OPT-B | $0.71 \pm 0.022$ | $1.0 \pm 0.3$ | $0.71 \pm 0.022$ | $1.0 \pm 0.3$ | $0.71 \pm 0.022$ | $1.1 \pm 0.3$ |
| 2OPT-LS | <NA> | <NA> | $0.696 \pm 0.013$ | $0.045 \pm 0.005$ | $0.693 \pm 0.013$ | $0.047 \pm 0.005$ |

Table 4: IJCNN: finding a tight solution: $(\epsilon_g, \epsilon_H) = (0.020, 0.141)$

| method | $\varepsilon = 0.2$ | | $\varepsilon = 0.6$ | | $\varepsilon = 1.0$ | |
| --- | --- | --- | --- | --- | --- | --- |
| | final loss | runtime | loss | runtime | loss | runtime |
| TR | $0.625 \pm 0.008$ | $\times$ | $0.59 \pm 0.009$ | $\times$ | $0.57 \pm 0.009$ | $\times$ |
| TR-B | $0.625 \pm 0.009$ | $\times$ | $0.591 \pm 0.009$ | $\times$ | $0.574 \pm 0.009$ | $\times$ |
| OPT | $0.643 \pm 0.011$ | $\times$ | $0.583 \pm 0.019$ | $\times$ | $0.515 \pm 0.039$ | $\times$ |
| OPT-B | $0.667 \pm 0.013$ | $1.1 \pm 0.3$ | $0.667 \pm 0.013$ | $0.9 \pm 0.1$ | $0.667 \pm 0.013$ | $1.0 \pm 0.1$ |
| OPT-LS | $0.835 \pm 0.069$ | $\times$ | $0.635 \pm 0.022$ | $\times$ | $0.598 \pm 0.024$ | $\times$ |
| 2OPT | $0.537 \pm 0.027$ | $\times$ | $0.522 \pm 0.074$ | $\times$ | $0.552 \pm 0.091$ | $\times$ |
| 2OPT-B | $0.666 \pm 0.014$ | $1.1 \pm 0.4$ | $0.666 \pm 0.014$ | $1.3 \pm 0.6$ | $0.666 \pm 0.014$ | $1.1 \pm 0.3$ |
| 2OPT-LS | <NA> | <NA> | $0.623 \pm 0.06$ | $\times$ | $0.649 \pm 0.004$ | $0.1 \pm 0.1$ |

We remark that for 2OPT-LS under $\varepsilon = 0.2$, the result is unavailable (reported as <NA>) because due to numerical issues, the package `autodp` we use cannot handle subsampling with a very low privacy budget.

### E.3  ADDITIONAL EXPERIMENTS

Additionally, we consider the logistic loss

$$\nabla l(w) = \frac{1}{n} \sum_{i=1}^{n} \frac{1}{1 + \exp\left(-y_i \langle x_i, w \rangle\right)} + \frac{\lambda}{2} \|w\|^2,$$

---

[5] Data source: LIBSVM data repository https://www.openml.org/search?type=data&sort=runs&id=1575&status=active

[6] Open source repo: https://github.com/yuxiangw/autodp

and repeat our experiments on the aforementioned datasets with $\lambda = 10^{-3}$. We can verify that the two chosen losses have Lipschitz gradients and Hessians as long as the feature vector $x_i$'s are bounded.

In this set of experiments, we find solutions $(\epsilon_g, \epsilon_H) = (0.040, 0.200)$ and $(\epsilon_g, \epsilon_H) = (0.020, 0.141)$. We also show the aggregated results for the number of noisy Hessian evaluations. We note that the number of noisy Hessian evaluations required in our algorithm is very low, whereas DP-TR needs to evaluate the noisy Hessian every iteration.

### E.3.1 COVERTYPE EXPERIMENT USING LOGISTIC LOSS

Table 5: Covertype (logistic loss): finding a loose solution: $(\epsilon_g, \epsilon_H) = (0.040, 0.200)$

| method | $\varepsilon = 0.2$ | | $\varepsilon = 0.6$ | | $\varepsilon = 1.0$ | |
|---|---|---|---|---|---|---|
| | final loss | runtime | loss | runtime | loss | runtime |
| TR | $0.425 \pm 0.009$ | $\times$ | $0.388 \pm 0.001$ | $\times$ | $0.381 \pm 0.001$ | $\times$ |
| TR-B | $0.425 \pm 0.009$ | $\times$ | $0.388 \pm 0.002$ | $\times$ | $0.382 \pm 0.001$ | $\times$ |
| OPT | $0.442 \pm 0.006$ | $\times$ | $0.539 \pm 0.022$ | $0.3 \pm 0.1$ | $0.539 \pm 0.022$ | $0.4 \pm 0.1$ |
| OPT-B | $0.539 \pm 0.022$ | $11.1 \pm 1.7$ | $0.539 \pm 0.022$ | $10.6 \pm 0.4$ | $0.539 \pm 0.022$ | $10.6 \pm 0.4$ |
| OPT-LS | $0.385 \pm 0.002$ | $\times$ | $0.455 \pm 0.014$ | $\times$ | $0.539 \pm 0.022$ | $0.4 \pm 0.1$ |
| 2OPT | $0.539 \pm 0.022$ | $0.3 \pm 0.1$ | $0.539 \pm 0.022$ | $0.3 \pm 0.1$ | $0.539 \pm 0.022$ | $0.4 \pm 0.1$ |
| 2OPT-B | $0.539 \pm 0.022$ | $1.0 \pm 0.0$ | $0.539 \pm 0.022$ | $1.0 \pm 0.1$ | $0.539 \pm 0.022$ | $1.0 \pm 0.1$ |
| 2OPT-LS | $0.539 \pm 0.022$ | $0.4 \pm 0.1$ | $0.539 \pm 0.022$ | $0.3 \pm 0.1$ | $0.539 \pm 0.022$ | $0.4 \pm 0.1$ |

Table 6: Covertype Hess evals (logistic loss): finding a loose solution: $(\epsilon_g, \epsilon_H) = (0.040, 0.200)$

| method | $\varepsilon = 0.2$ | | $\varepsilon = 0.6$ | | $\varepsilon = 1.0$ | |
|---|---|---|---|---|---|---|
| | Hess evals | runtime | Hess evals | runtime | Hess evals | runtime |
| TR | $375.0 \pm 0.0$ | $\times$ | $375.0 \pm 0.0$ | $\times$ | $375.0 \pm 0.0$ | $\times$ |
| TR-B | $375.0 \pm 0.0$ | $\times$ | $375.0 \pm 0.0$ | $\times$ | $375.0 \pm 0.0$ | $\times$ |
| OPT | $190.6 \pm 32.3$ | $\times$ | $1.0 \pm 0.0$ | $0.3 \pm 0.1$ | $1.0 \pm 0.0$ | $0.4 \pm 0.1$ |
| OPT-B | $1.0 \pm 0.0$ | $11.1 \pm 1.7$ | $1.0 \pm 0.0$ | $10.6 \pm 0.4$ | $1.0 \pm 0.0$ | $10.6 \pm 0.4$ |
| OPT-LS | $0.0 \pm 0.0$ | $\times$ | $282.4 \pm 174.252$ | $\times$ | $1.0 \pm 0.0$ | $0.4 \pm 0.1$ |
| 2OPT | $1.0 \pm 0.0$ | $0.3 \pm 0.1$ | $1.0 \pm 0.0$ | $0.3 \pm 0.1$ | $1.0 \pm 0.0$ | $0.4 \pm 0.1$ |
| 2OPT-B | $1.0 \pm 0.0$ | $1.0 \pm 0.0$ | $1.0 \pm 0.0$ | $1.0 \pm 0.1$ | $1.0 \pm 0.0$ | $1.0 \pm 0.1$ |
| 2OPT-LS | $1.0 \pm 0.0$ | $0.4 \pm 0.1$ | $1.0 \pm 0.0$ | $0.3 \pm 0.1$ | $1.0 \pm 0.0$ | $0.4 \pm 0.1$ |

Table 7: Covertype (logistic loss): finding a tight solution: $(\epsilon_g, \epsilon_H) = (0.020, 0.141)$

| method | $\varepsilon = 0.2$ | | $\varepsilon = 0.6$ | | $\varepsilon = 1.0$ | |
|---|---|---|---|---|---|---|
| | final loss | runtime | loss | runtime | loss | runtime |
| TR | $0.408 \pm 0.004$ | $\times$ | $0.381 \pm 0.0$ | $\times$ | $0.378 \pm 0.0$ | $\times$ |
| TR-B | $0.408 \pm 0.004$ | $\times$ | $0.381 \pm 0.0$ | $\times$ | $0.378 \pm 0.0$ | $\times$ |
| OPT | $0.379 \pm 0.001$ | $\times$ | $0.38 \pm 0.001$ | $\times$ | $0.39 \pm 0.002$ | $\times$ |
| OPT-B | $0.454 \pm 0.004$ | $1.3 \pm 0.2$ | $0.454 \pm 0.004$ | $1.4 \pm 0.4$ | $0.454 \pm 0.004$ | $1.5 \pm 0.6$ |
| OPT-LS | $0.41 \pm 0.005$ | $\times$ | $0.381 \pm 0.001$ | $\times$ | $0.378 \pm 0.0$ | $\times$ |
| 2OPT | $0.386 \pm 0.006$ | $\times$ | $0.378 \pm 0.0$ | $\times$ | $0.377 \pm 0.0$ | $\times$ |
| 2OPT-B | $0.454 \pm 0.004$ | $2.0 \pm 0.2$ | $0.454 \pm 0.004$ | $2.0 \pm 0.3$ | $0.454 \pm 0.004$ | $1.8 \pm 0.2$ |
| 2OPT-LS | $0.441 \pm 0.007$ | $\times$ | $0.447 \pm 0.009$ | $0.6 \pm 0.2$ | $0.447 \pm 0.008$ | $0.7 \pm 0.2$ |

Table 8: Covertype Hess evals (logistic loss): finding a tight solution: $(\epsilon_g, \epsilon_H) = (0.020, 0.141)$

| method | $\varepsilon = 0.2$ | | $\varepsilon = 0.6$ | | $\varepsilon = 1.0$ | |
|---|---|---|---|---|---|---|
| | Hess evals | runtime | Hess evals | runtime | Hess evals | runtime |
| TR | $1061.0 \pm 0.0$ | $\times$ | $1061.0 \pm 0.0$ | $\times$ | $1061.0 \pm 0.0$ | $\times$ |
| TR-B | $1061.0 \pm 0.0$ | $\times$ | $1061.0 \pm 0.0$ | $\times$ | $1061.0 \pm 0.0$ | $\times$ |
| OPT | $0.0 \pm 0.0$ | $\times$ | $58.6 \pm 24.765$ | $\times$ | $372.6 \pm 141.077$ | $\times$ |
| OPT-B | $1.0 \pm 0.0$ | $1.3 \pm 0.2$ | $1.0 \pm 0.0$ | $1.4 \pm 0.4$ | $1.0 \pm 0.0$ | $1.5 \pm 0.6$ |
| OPT-LS | $0.0 \pm 0.0$ | $\times$ | $0.0 \pm 0.0$ | $\times$ | $1.0 \pm 1.0$ | $\times$ |
| 2OPT | $0.0 \pm 0.0$ | $\times$ | $0.0 \pm 0.0$ | $\times$ | $0.2 \pm 0.447$ | $\times$ |
| 2OPT-B | $1.0 \pm 0.0$ | $2.0 \pm 0.2$ | $1.0 \pm 0.0$ | $2.0 \pm 0.3$ | $1.0 \pm 0.0$ | $1.8 \pm 0.2$ |
| 2OPT-LS | $33.0 \pm 6.745$ | $\times$ | $1.0 \pm 0.0$ | $0.6 \pm 0.2$ | $1.0 \pm 0.0$ | $0.7 \pm 0.2$ |

### E.3.2 IJCNN EXPERIMENT USING LOGISTIC LOSS

Table 9: IJCNN (logistic loss): finding a loose solution: $(\epsilon_g, \epsilon_H) = (0.040, 0.200)$

| method | $\varepsilon = 0.2$ | | $\varepsilon = 0.6$ | | $\varepsilon = 1.0$ | |
|---|---|---|---|---|---|---|
| | final loss | runtime | loss | runtime | loss | runtime |
| TR | $0.477 \pm 0.01$ | $\times$ | $0.454 \pm 0.006$ | $\times$ | $0.446 \pm 0.004$ | $\times$ |
| TR-B | $0.477 \pm 0.01$ | $\times$ | $0.454 \pm 0.006$ | $\times$ | $0.446 \pm 0.004$ | $\times$ |
| OPT | $0.46 \pm 0.01$ | $\times$ | $0.439 \pm 0.002$ | $\times$ | $0.463 \pm 0.005$ | $\times$ |
| OPT-B | $0.501 \pm 0.008$ | $8.6 \pm 0.2$ | $0.501 \pm 0.008$ | $11.0 \pm 4.9$ | $0.501 \pm 0.008$ | $9.0 \pm 1.0$ |
| OPT-LS | $0.607 \pm 0.076$ | $\times$ | $0.473 \pm 0.012$ | $\times$ | $0.454 \pm 0.004$ | $\times$ |
| 2OPT | $0.493 \pm 0.022$ | $\times$ | $0.501 \pm 0.008$ | $0.0^7 \pm 0.0$ | $0.501 \pm 0.008$ | $0.0 \pm 0.0$ |
| 2OPT-B | $0.501 \pm 0.008$ | $1.0 \pm 0.0$ | $0.501 \pm 0.008$ | $1.4 \pm 0.9$ | $0.501 \pm 0.008$ | $1.1 \pm 0.3$ |
| 2OPT-LS | $1.002 \pm 0.321$ | $\times$ | $0.501 \pm 0.008$ | $0.0 \pm 0.0$ | $0.501 \pm 0.008$ | $0.0 \pm 0.0$ |

Table 10: IJCNN Hess evals (logistic loss): finding a loose solution: $(\epsilon_g, \epsilon_H) = (0.040, 0.200)$

| method | $\varepsilon = 0.2$ | | $\varepsilon = 0.6$ | | $\varepsilon = 1.0$ | |
|---|---|---|---|---|---|---|
| | Hess evals | runtime | Hess evals | runtime | Hess evals | runtime |
| TR | $375.0 \pm 0.0$ | $\times$ | $375.0 \pm 0.0$ | $\times$ | $375.0 \pm 0.0$ | $\times$ |
| TR-B | $375.0 \pm 0.0$ | $\times$ | $375.0 \pm 0.0$ | $\times$ | $375.0 \pm 0.0$ | $\times$ |
| OPT | $0.0 \pm 0.0$ | $\times$ | $0.5 \pm 0.577$ | $\times$ | $85.25 \pm 47.647$ | $\times$ |
| OPT-B | $1.0 \pm 0.0$ | $8.6 \pm 0.2$ | $1.0 \pm 0.0$ | $11.0 \pm 4.9$ | $1.0 \pm 0.0$ | $9.0 \pm 1.0$ |
| OPT-LS | $0.0 \pm 0.0$ | $\times$ | $0.0 \pm 0.0$ | $\times$ | $0.25 \pm 0.5$ | $\times$ |
| 2OPT | $6.5 \pm 1.732$ | $\times$ | $1.0 \pm 0.0$ | $0.0 \pm 0.0$ | $1.0 \pm 0.0$ | $0.0 \pm 0.0$ |
| 2OPT-B | $1.0 \pm 0.0$ | $1.0 \pm 0.0$ | $1.0 \pm 0.0$ | $1.4 \pm 0.9$ | $1.0 \pm 0.0$ | $1.1 \pm 0.3$ |
| 2OPT-LS | $0.25 \pm 0.5$ | $\times$ | $1.0 \pm 0.0$ | $0.0 \pm 0.0$ | $1.0 \pm 0.0$ | $0.0 \pm 0.0$ |

Table 11: IJCNN (logistic loss): finding a tight solution: $(\epsilon_g, \epsilon_H) = (0.020, 0.141)$

| method | $\varepsilon = 0.2$ | | $\varepsilon = 0.6$ | | $\varepsilon = 1.0$ | |
|---|---|---|---|---|---|---|
| | final loss | runtime | loss | runtime | loss | runtime |
| TR | $0.478 \pm 0.01$ | $\times$ | $0.454 \pm 0.005$ | $\times$ | $0.446 \pm 0.003$ | $\times$ |
| TR-B | $0.479 \pm 0.01$ | $\times$ | $0.454 \pm 0.005$ | $\times$ | $0.446 \pm 0.003$ | $\times$ |
| OPT | $0.484 \pm 0.012$ | $\times$ | $0.447 \pm 0.007$ | $\times$ | $0.44 \pm 0.003$ | $\times$ |
| OPT-B | $0.501 \pm 0.007$ | $7.9 \pm 6.3$ | $0.501 \pm 0.007$ | $7.8 \pm 6.2$ | $0.501 \pm 0.007$ | $9.2 \pm 7.4$ |
| OPT-LS | $0.761 \pm 0.066$ | $\times$ | $0.553 \pm 0.037$ | $\times$ | $0.492 \pm 0.016$ | $\times$ |
| 2OPT | $0.527 \pm 0.024$ | $\times$ | $0.462 \pm 0.013$ | $\times$ | $0.501 \pm 0.007$ | $0.0 \pm 0.0$ |
| 2OPT-B | $0.502 \pm 0.007$ | $1.5 \pm 0.4$ | $0.502 \pm 0.007$ | $1.3 \pm 0.1$ | $0.502 \pm 0.007$ | $1.6 \pm 0.3$ |
| 2OPT-LS | $3.504 \pm 0.98$ | $\times$ | $0.798 \pm 0.081$ | $\times$ | $0.459 \pm 0.01$ | $0.4 \pm 0.4$ |

Table 12: IJCNN Hess evals (logistic loss): finding a tight solution: $(\epsilon_g, \epsilon_H) = (0.020, 0.141)$

| method | $\varepsilon = 0.2$ | | $\varepsilon = 0.6$ | | $\varepsilon = 1.0$ | |
|---|---|---|---|---|---|---|
| | Hess evals | runtime | Hess evals | runtime | Hess evals | runtime |
| TR | $1061.0 \pm 0.0$ | $\times$ | $1061.0 \pm 0.0$ | $\times$ | $1061.0 \pm 0.0$ | $\times$ |
| TR-B | $1061.0 \pm 0.0$ | $\times$ | $1061.0 \pm 0.0$ | $\times$ | $1061.0 \pm 0.0$ | $\times$ |
| OPT | $0.0 \pm 0.0$ | $\times$ | $0.0 \pm 0.0$ | $\times$ | $0.0 \pm 0.0$ | $\times$ |
| OPT-B | $1.0 \pm 0.0$ | $7.9 \pm 6.3$ | $1.0 \pm 0.0$ | $7.8 \pm 6.2$ | $1.0 \pm 0.0$ | $9.2 \pm 7.4$ |
| OPT-LS | $0.0 \pm 0.0$ | $\times$ | $0.0 \pm 0.0$ | $\times$ | $0.0 \pm 0.0$ | $\times$ |
| 2OPT | $0.0 \pm 0.0$ | $\times$ | $3.0 \pm 1.581$ | $\times$ | $1.0 \pm 0.0$ | $0.0 \pm 0.0$ |
| 2OPT-B | $1.0 \pm 0.0$ | $1.5 \pm 0.4$ | $1.0 \pm 0.0$ | $1.3 \pm 0.1$ | $1.0 \pm 0.0$ | $1.6 \pm 0.3$ |
| 2OPT-LS | $0.0 \pm 0.0$ | $\times$ | $0.4 \pm 0.548$ | $\times$ | $4.8 \pm 4.97$ | $0.4 \pm 0.4$ |

