# OpenReview forum: "Differentially Private Algorithms for Smooth Nonconvex ERM"
_ICLR.cc/2023/Conference — Submitted to ICLR 2023_

### Official Review · Reviewer_CxW8 · 2022-10-18

**Confidence:** 4
**Correctness:** 4
**Technical Novelty And Significance:** 3
**Empirical Novelty And Significance:** 3
**Recommendation:** 6

**Clarity, Quality, Novelty And Reproducibility:**

The paper is largely clearly written. However, the motivation of the algorithms are not clear. It would be helpful if the authors can give some explanation of the algorithm.

The paper does not use the standard template of ICLR2023. The authors need to address this in the revised version.

There is a gap between the derived sample complexity and the existing bound. It would be helpful if the authors can explain the challenge in bridging this gap.

Above Eq (7), the authors say the step sizes are independent of k. However, this is not the case according to the step size in Eq (7):

Above Lemma 2: "oef" should be "of"

Section 4: "columns is" should be "columns are"

Proof of Theorem 8: "similar to D.4" should be "similar to Corollary D.4"?

The section of conclusion is not provided.

Definition A.2: the inequality should be an equality

Definition A.4: it seems that "$M$ satisfies $(\alpha,\xi+\rho\alpha)$-RDP" is not correct

Proposition A.6: "$(\epsilon,\delta)$-DP to zCDP" should be "zCDP to $(\epsilon,\delta)$-DP"

Proposition A.7: $\Delta_f$ should be $\Delta_h$

Below Eq (30): there is ??

**Strength And Weaknesses:**

**Strength**

Most of the DP algorithms focus on finding first-order approximate solutions of ERM problems. This paper aims to find second-order approximate solutions, which are much more challenging.

Both theoretical and experimental analysis are presented for the proposed algorithm with and without line search.

**Weakness**

The paper only discusses results on sample complexity. There is no analysis on the computation complexity for the proposed algorithm, which is also an important issue.

The sample complexity is of the order $O(\alpha^{-2})$, which is worse than the state of the art $O(\alpha^{-7/4})$.

The theoretical analysis assumes a bounded gradient assumption on the loss function. In the experiments, the function involves a regularizer $\sum_i\lambda w_i^2/(1+w_i^2)$, whose gradients are not bounded. Therefore, the theoretical analysis and experimental analysis are not consistent.

**Summary Of The Paper:**

This paper presents differentially private algorithms to find approximate solutions with second-order guarantees for nonconvex problems, i.e., solutions with small gradients and almost positive definite Hessian matrices. The paper first develops an algorithm with step size depending on some parameters of problems, and then extends it to the algorithms with line search. A minibatch variant is also discussed. Experimental results are also presented to show the effectiveness of the proposed algorithm.


**Summary Of The Review:**

The paper considers a challenging problem of finding the second-order approximate solutions for smooth ERM problems. The analysis seems to be challenging and interesting. There is a gap between theory and experiments: the theoretical analysis requires a bounded gradient assumption, which does not hold for the problem considered in the experiments.

---

> ### Author Response · Authors · 2022-11-19
> **Response to Reviewer CxW8**
>
> Thanks for your detailed reading and critique!
>
> > The paper only discusses results on sample complexity. There is no analysis on the computation complexity for the proposed algorithm, which is also an important issue.
> >
>
> The iteration complexity for our algorithms is captured in the quantity $T$, the upper bound on the number of iterations. Since $T$ is inversely proportional to MIN_DEC, we have $T=O(\max(\epsilon_g^{-2},\epsilon_H^{-3}))$. Each iteration of our method requires a (noisy) gradient evaluation. On iterations on which the gradient norm falls below the threshold specified to qualify as an approximate first-order optimal point, the Hessian is also evaluated and its minimum eigenvalue and eigenvector is determined. The cost of the latter operation depend on how it is performed - if by a randomized Lanczos process, it costs $O(\epsilon_H^{-1/2})$ Hessian-vector multiplications. By contrast, DP-GD requires $O(\epsilon_g^{-2})$ gradient evaluations, and can be implemented in a way that the Hessian evaluations can be required only at those points at which the gradient norm falls below the threshold. (For each Hessian they require an eigenvalue calculation similar to ours.) If we set $(\epsilon_g,\epsilon_H)=(\alpha,\sqrt{M\alpha})$, then these complexities are both $O(\alpha^{-2})$. By contrast, for a similar setting of optimality tolerances, DP-TR will has an iteration complexity of $O(\alpha^{-3/2})$. However, DP-TR requires calculation of the Hessian on every iteration as well as exact solution of a trust-region subproblem, an operation whose implementation is not specified but which is typically more complex than an eigenvalue calculation. If the complexity of this operation is assumed to be the same as randomized Lanczos for extreme eigenvalue calculation - $O(\alpha^{-1/4})$ - then the computational complexity of DP-TR would be $O(\alpha^{-7/4})$, whereas our method has a computational complexity of $O(\alpha^{-2})$ when randomized Lanczos is used. *SW: Should we also discuss the effects on iteration complexity of minibatching?*
>
> > The theoretical analysis assumes a bounded gradient assumption on the loss function. In the experiments, the function involves a regularizer $\sum_i \frac{\lambda w_i^2}{1+w_i^2}$, whose gradients are not bounded. Therefore, the theoretical analysis and experimental analysis are not consistent.
> >
>
> The regularizer $r(w)=\frac{w^2}{1+w^2}$ has bounded gradients and Hessians. Its gradient is $r'(w)=\frac{2w}{(1+w^2)^2}$, and by using the inequality $1+w^2 \ge 2 |w|$, we have $|r'(w)| \le \frac{1}{1+w^2} \le 1$, for all $w$.  (A more refined analysis gives an upper bound of $3 \sqrt{3}/8$.) For the Hessian, an elementary calculation shows that $|r’’(w)|$ attains its maximum value at $w=0$, where $r’’(0)=2$.
>
> > The paper does not use the standard template of ICLR2023. The authors need to address this in the revised version.
> >
>
> We are using the ICLR template. Can you point out which part(s) has format issues?
> > Definition A.4: it seems that "M satisfies (α,ξ+ρα)-RDP" is not correct
> >
>
> Can you elaborate why this is incorrect?
>
> All other typos were fixed in the revision. Thank you for pointing them out.

---

> > ### Comment · Reviewer_CxW8 · 2022-11-22
> > **Thank you for your response**
> >
> > Yes. You are right. The gradient and the Hessian for the regularizer are bounded. Sorry for my misunderstanding.
> >
> > For the standard ICLR submission, I think there should be "Under review as a conference paper at ICLR 2023" in the header.

---

### Official Review · Reviewer_LsJR · 2022-10-22

**Confidence:** 4
**Correctness:** 3
**Technical Novelty And Significance:** 2
**Empirical Novelty And Significance:** 2
**Recommendation:** 3

**Clarity, Quality, Novelty And Reproducibility:**

The presentation is clear. However, it is unclear how to keep track of the privacy loss and determine $\sigma,\sigma_g,\sigma_H$  when we implement the algorithm in practice.

**Strength And Weaknesses:**

The strength of the paper:
1. The proposed method is able to find the 2NS of the nonconvex ERM with privacy guarantees.
2. The convergence guarantee is established.
3. A line search-based algorithm is developed, which could be of independent interest.

The Weaknesses of the paper:
1. The proposed method seems to be a straightforward extension of the existing nonprivate method by injecting random Gaussian noise to objectives, gradients, and Hessians. What is the key challenge when you derive the theoretical guarantees compared with the nonprivate counterpart?
2. To control the error introduced by the random noise, the proposed method require a stringent condition on the sample complexity. This condition could be violated in practice. Therefore, a more meaningful question could be: given a fixed sample size $n$ and the privacy budget $\rho$, what is the best $(\epsilon_{g},\epsilon_{H})$-2NS one can achieve using your proposed method.
3. The proposed method needs to compute the Hessian matrix, which could be time-consuming.
4. It seems that there is no advantage in terms of sample complexity when we use line search-based algorithm.
5. A single experiment with a small dataset is not enough to validate the effectiveness of the proposed method. The authors should consider experiments using neural networks.

**Summary Of The Paper:**

This paper studies the problem of differentially private nonconvex ERM. More specifically, the authors propose a differentially private method aiming to find the approximate second-order necessary solution. The proposed method is a straightforward extension of the existing nonprivate method by injecting random Gaussian noise into the objectives, gradients, and Hessians. The authors provide the convergence guarantee of the proposed method, and an experiment validates the effectiveness of the proposed method.

**Summary Of The Review:**

The problem considered in this paper is very relevant and interesting. However, the proposed method seems to be incremental. In addition, the requirement for the sample seems to be very strong, and it is unclear how to determine the privacy related parameters in practice.

---

> ### Author Response · Authors · 2022-11-19
> **Response to Reviewer LsJR**
>
> We thank the reviewer for the comments and address the concerns below.
>
> Regarding the weakness of the paper,
>
> 1. We argue that it is an *advantage* for our algorithm to have a simpler, more transparent analysis. DP-GD achieves the same rate with a more complicated analysis. In any case, the analysis of the line search in our algorithm is much more technical than the analysis of backtracking line search in a non-private algorithm.
> 2. The result described by the referee could be obtained by inverting the sample complexity formula. Specifically, in Theorem 4, the sample complexity $n_{\min}$ could be fixed, substitutions $\epsilon_g=\alpha$ and $\epsilon_H = \sqrt{M \alpha}$ made, and the resulting equation solved for $\alpha$. (Note that $T$ depends on $\alpha$, but since $T$ enters only logarithmically into the formula, this induces only a minor complication.)
> 3. To our knowledge, DP methods so far for finding a 2NS all require the use of the Hessian. This claim includes DP-GD, in which all the steps taken are gradient steps, but Hessians are evaluated to check for the second-order condition, and DP-TR, in which Hessians are required at every step. By comparison with other methods, ours is more sparing in its use of Hessian information (making use of the Hessian only when the noisy gradient norm falls below its convergence threshold), and makes better use of the Hessian information (using a direction of negative curvature to compute a descent step for the function). Empirically, we see in our numerical tests that the Hessian is required on relatively few iterations.
> 4. It is common, perhaps universal, in (non-private) optimization algorithms that the use of line searches does not improve worst-case complexity over the use of safe, short steps at every iteration. But methods that make use of line search and similar features are still useful because (a) they no longer assume the detailed knowledge of the functions that define the problem that is needed to set the short steplengths appropriately; (b) they are less conservative than the short-step approaches and can adapt to the local geometry of the problem, usually leading to better computational performance. These facts provide the motivation for the line-search variant of our private algorithm. The practicality of this variant is demonstrated in our experiments.
> 5. We added additional experiments in our revision. See Appendix.
>
> > it is unclear how to keep track of the privacy loss and determine $\sigma, \sigma_g, \sigma_H$ when we implement the algorithm in practice.
> >
>
> We stated the choice of $\sigma, \sigma_g, \sigma_H$ in our theorems. Track of privacy loss is done by applying the composition theorems of differential privacy. To deal with subsampling complexities, we use RDP accounting tool `autodp`.
>
> The relationships that we derive between privacy, accuracy, and sample size are indeed “strong” as they are based on conservative assumptions about the function properties. The other features that we introduce in this paper, including the use of line search and the two-phase approach, give us the opportunity to obtain much better performance in practice.
>
> > A few statements are not well-supported, or require small changes to be made correct.
> >
>
> If you could point these out, we would be happy to fix them. (We have tried to check for such statements in an additional scan.)

---

### Official Review · Reviewer_8RwJ · 2022-10-27

**Confidence:** 2
**Correctness:** 4
**Technical Novelty And Significance:** 3
**Empirical Novelty And Significance:** 2
**Recommendation:** 6

**Clarity, Quality, Novelty And Reproducibility:**

Clarity: this paper is in general well-written.

Quality: the results provided by the paper seem correct

Novelty: the proposed algorithm is novel.

Reproducibility: proofs for the theorems seem correct.

**Strength And Weaknesses:**

Strength:
 The problem studied in the paper is important and interesting. The proposed algorithm makes sense.

Weaknesses:

1. The presentation of the paper could be further improved. For instance, the benefits of using nonconvex loss functions should be clarified in the introduction section. In Assumptions 1 and 2, restrictions on loss functions are imposed. Further remarks, including loss functions that satisfy the assumptions, are expected here.

2. I think more details regarding the numerical experiments should be provided. For example, how the tuning parameters were selected is not clear to me. And it is also not clear why this particular loss function is chosen in numerical experiments. I expect more extensive simulation results.



**Summary Of The Paper:**

In this paper, the authors proposed differentially private optimization algorithms for empirical risk minimization. To improve the speed and practicality of the algorithm, the authors proposed to use several simple yet effective strategies such as line search and mini-batching. The effectiveness of the proposed approach is validated through numerical experiments.

**Summary Of The Review:**

see above

---

> ### Author Response · Authors · 2022-11-19
> **Response to Reviewer 8RwJ**
>
> We thank the reviewer for the comments. Below we address the concerns.
>
> Weakness #1: Nonconvex loss functions are useful in this finite-sum context in the setting of robust statistics. For example, in a linear regression problem, when for certain distributions of noise in observations, a loss function like Huber’s function or the Tukey biweight function can be used to reduce sensitivity to outliers. Nonconvexity also appears in the regularization terms in robust statistics, via such regularizers as MCP and SCAD. We also include additional experiments using logistic loss and $l^2$ regularizer in our revised paper.
>
> Weakness #2: Conventional hyperparameter optimization can be difficult to perform on a case-by-case basis in this setting because of the loss of privacy it would entail. Although our line-search method (Algorithms 3 and 4) contain many parameters, most of them have fairly obvious values and performance can be expected to be robust to these values. For example, the parameters $b_g$, $b_H$, $\beta_g$, and $\beta_H$ associated with the line search and the sufficient decrease parameters $c_g$ and $c_H$ have “conventional wisdom” values in optimization. Appropriate settings of the inexactness values $c_1$, $c_2$, $c$ are clear from the analysis. See Appendix in our revised paper for more numerical results.

---

### Official Review · Reviewer_2L65 · 2022-10-31

**Confidence:** 3
**Correctness:** 3
**Technical Novelty And Significance:** 2
**Empirical Novelty And Significance:** 2
**Recommendation:** 5

**Clarity, Quality, Novelty And Reproducibility:**

For the most part the paper is clearly written and provides adequate details in the proof, although the experiments sections seems to lack a comparison to the DP-GD method of (Wang et al, 2019). I do think that the proof of Theorem 1 should make a comment with regards to the sensitivity bound. There are a few minor typos/broken references that could easily be fixed.

**Strength And Weaknesses:**

Strengths:

The paper provides sound and precise proofs and theorem statements. The paper provides both a simple and more involved (line search) version of their algorithm, which makes the overall analysis easier to follow and provides a deeper look into their method. The authors provide experiments to support the rational for using line search, and show that this method leads to faster running time.

Weaknesses:

The results of the paper may be of limited interest. The rates achieved by the algorithm for obtaining a second order stationary point do not improve upon existing rates and are in fact worse than DPTR as the authors themselves note. The authors argue that their algorithm is more practical than DPTR, because less Hessian evaluations are needed. While this is true, the algorithm seems no more practical than the the DP-GD algorithm of (Wang et al., 2019), as there too the Hessian would only need to be evaluated after first order stationarity is checked. Further, DP-GD achieves the same rate. The authors do propose a line search variant to further improve the practical running time, but the fact that any Hessian computations are still needed limits the reach of this improvement.

**Summary Of The Paper:**

The paper provides formal guarantees for finding second order stationary points of nonconvex loss functions under differential privacy and certain (standard) smoothness assumptions on the loss function. The paper provides a formal analysis for a noisy version of the "gradient descent with negative curvature steps" algorithm. Privacy is ensured via noisy versions of the gradient or Hessian at each step. The paper also analyzes a DP line search (for step sizes) variant of the algorithm which guarantees privacy via the sparse vector technique. Finally, the paper provides experiments for a nonconvex loss function to study the effectiveness of their method.

**Summary Of The Review:**

The paper is well written and proposes a new algorithm/analysis for finding second order stationary points. However, the results obtained by this analysis do not improve over existing methods.

---

> ### Author Response · Authors · 2022-11-19
> **Response to Reviewer 2L65**
>
> Thank you for the comment. We address your concerns below.
>
> Regarding the *theoretical* comparison to DP-GD, we reiterate that (see the introduction section of our paper) "DP-GD uses the (noisy) Hessian only for checking the second-order approximate condition”. First, we make more efficient use of the (noisy) Hessian—we move along the most negative curvature direction of this matrix. Specifically, DP-GD requires $\Gamma = O(\alpha^{-1/2})$ steps to obtain an $O(\alpha^{-3/2})$ decrease in the function. By contrast, our method achieves the same order of decrease in just one negative curvature step (if the minimum-eigenvalue calculation is done by a direct eigenvalue decomposition method) or $O(\alpha^{-1/4})$ steps (if the randomized Lanczos method of Appendix C is used with $\epsilon= \epsilon_H = O(\alpha^{1/2})$). Second, we evaluate the Hessian only when the norm of the (noisy) gradient falls below the threshold required to declare the point approximately optimal. Third, in the original DP-GD, gradients are evaluated a fixed number of times, whereas our method often stops early. We also note that our analysis is much more straightforward than that of DP-GD.
>
> Regarding *practical* comparisons with DP-GP and DP-TR, we note that these two prior algorithms are based on conservative assumptions, concerning the global geometry of the problem, which lead to very small choices of steplengths / trust region radii and very large added noise, and thus very slow convergence. Our main goal in this paper is to propose approaches that look more like *practical* algorithms, that can be expected to find near-optimal points privately and efficiently, in a wider range of scenarios than before, while still having theoretical performance guarantees (regarding sample complexity and computational complexity) that are close to the state of the art. Accordingly, we have introduced two algorithmic features - a line search and a two-phase approach - which, along with minibatching, greatly enhance practicality without sacrificing theoretical performance guarantees. We use the privacy measure  $\rho$-zCDP which can be monitored iteration-by-iteration to keep track of privacy loss as the algorithm proceeds. For minibatching, we use RDP since it is capable of handling the privacy accounting for subsampling.
>
> As noted in the revised paper, we were not able to do a direct comparison with DP-GD. When we tried to set algorithmic parameters as described in the DP-GD paper, the values were inconsistent with reasonable practical behavior. Results for DP-GD are reported in the DP-TR paper, but no details are given in that paper as to parameter settings, so we cannot reproduce these results. We have emailed the author for implementation details and will follow up on this. We note that in the latter paper, results for DP-GD and DP-TR are not vastly different, so we believe that a comparison of our approaches with DP-TR suffice to establish its possible interest as a practical approach.
>
> > I do think that the proof of Theorem 1 should make a comment with regards to the sensitivity bound.
>
> Are you suggesting we should comment that the noise added is already scaled by the sensitivity? We added a remark after the theorem.

---

### Author Response · Authors · 2022-11-19
**Paper Revision**

Thanks for reviewers' helpful comments. GIven the feedback, we have revised our paper and colored significant changes blue.
Below is the **What's New** for the revision:
- Revamp the experiments section. Improve the visuals and give more explanation of the results.
- Comment that only Hessian vector product is needed for Lanczos method. The evaluation of the full Hessian is not necessary. Plus, the number of Hessian evaluations is minimal, as shown in the experiments.
- Add additional experiments in the Appendix.
- Other minor fixes.

---

> ### Comment · Reviewer_2L65 · 2022-11-21
> **Hessian vector products**
>
> Thank you for your revisions. However, it is unclear to me how a method like finite differencing would allow you to compute Hessian vector products of the \textit{noisy} Hessian, and no details are provided in the revision. Also, if the final result is only an approximation of the minimum eigenvector of the noisy Hessian, and that approximation is computed using data dependent quantities, it is not even clear that the result is still private.

---

> > ### Author Response · Authors · 2022-11-24
> > **Response to Reviewer 2L65**
> >
> > ##
> >
> > Thank you for the comment. While finite differencing only gives an approximation, you can still add Gaussian noise to it. Let $v$ be a unit vector and
> >
> > $$
> > \alpha \cdot hvp = \|\nabla f(w+\alpha v) - \nabla f(w)\|.
> > $$
> >
> > Using the $M$-lipschitzness of the gradient, we can compute the sensitivity by (let $x_k$ and $x_k'$ be the different elements of two neighboring datasets)
> >
> > $$
> > \frac1n\left(\|\nabla l(w+\alpha v, x_k) - \nabla l(w, x_k) \| + \|\nabla l(w+\alpha v, x_k') - \nabla l(w, x_k')\|\right) \\
> > \le \frac2n M \alpha \|v\| = \frac2n M \alpha.
> > $$
> >
> > Therefore, the sensitivity of $hvp$ is bounded by $\frac2n M$ and we can use Gaussian mechanism to privative the finite difference. You will have to do this for every Hessian-vector product in the Lanczos iteration. But considering we normally only take a tiny number of curvature steps, this can still be acceptable.
> >
> > A better way is to use automatic differentiation as mentioned in the paper before “finite differencing”. In that case, given the Gaussian noise matrix $E$, for every direction $v$, we run automatic differentiation with respect to $\nabla f(w)^Tv$ and obtain $Hv$. We then add $Ev$ to it; effectively we are still computing $(H+E)v$, albeit using automatic differentiation to save some computation.
> >
> > That being said, as we mentioned (also see the additional tables in the appendix) we typically only need a few Hessian evaluations, so the cost could be acceptable. We will add the details above to our final version.

---

### Author Response · Authors · 2022-12-04
**Reminder From Authors**

Thanks again for the valuable feedback from all reviewers.
Meanwhile, the deadline for Discussion Stage 2 is approaching. Please kindly let us know if we have addressed your concerns. We will be happy to respond to additional comments.

---

### Decision · Program_Chairs · 2023-01-20

**Decision:**

Reject

**Justification For Why Not Higher Score:**

The paper have several major shortcomings. The largest is that it does not provide any clear improvements over prior work.

**Justification For Why Not Lower Score:**

N/A

**Metareview: Summary, Strengths And Weaknesses:**

This paper formally studies differentially private algorithms for approximating second-order stationary points of nonconvex losses.

The authors give formal convergence guarantees for noisy SGD with negative curvature search. The authors also propose a differentially private line search method for selecting the step size. The theoretical results are complemented with numerical experiments to test the performance of the proposed methods.

Although the authors provide rigorous analyses and formal guarantees for a fundamental problem of considerable importance to the area of privacy-preserving optimization, the results are mostly incremental. The formal results do not provide rate improvements for this problem compared to prior constructions (DP-TR [Wang & Xu 2021] and DP-GD [Wang et al.  2019]).

The claims of practicality of the proposed methods compared to the existing ones are not very convincing due to the lack of formal, precise statements showing a quantifiable computational advantage over the prior methods (particularly, the authors do not provide precise discussion of the computational cost of the noisy Hessian and the noisy Hessian-vector approximation and also the impact of the approximation on the convergence rate).

The proposed methods are also of limited novelty as they are natural and obvious  extensions of existing methods in the literature of non-convex optimization. There are also some concerns about the correctness of the claim that $O(d^2)$ running time can be avoided. The argument provided for that claim is not precise and sufficiently detailed.

The above shortcomings make the work of limited interest.